# Multi-layered transcriptional control of glycogen metabolism coordinates thermogenic remodeling of white adipocytes in male mice

Haipeng Fu [1], Seoyeon Lee[2,6], Nathan R. Zemke[2,3], Weiwei Fan[4], Yunqing Wang[1], James Garza[1], David Tin[1], Bryce Villao[1], Bichen Zhang[1], Xianda Ma[1], Jinyang Zhang[1], Tangran Dong[1], Yuyao Ren[1], Michael Downes[4], Ronald M. Evans [4], Bing Ren[2,3,5,7] & Alan R. Saltiel [1] ✉

Thermogenic activation of subcutaneous white adipocytes requires glycogen synthesis and turnover. Here we show that β-adrenergic stimulation induces a distinct glycogen metabolism gene program in inguinal white adipose tissue in a cell-autonomous and adipocyte-specific manner. Among these, *Gys2* and *Ppp1r3c* are rapidly induced following acute β3-adrenergic receptor activation. We identify *Gys2* as a direct transcriptional target of PKA-CREB signaling. In contrast, sustained expression of glycogen metabolism genes under chronic β3-adrenergic activation requires the coactivator PGC1α, whose loss blunts glycogen accumulation and thermogenic capacity. Mechanistically, PGC1α cooperates with estrogen-related receptors (ERRs) to regulate chromatin accessibility and gene transcription. Although deletion of ERRα is compensated by ERRγ, combined deletion of ERRα/β/γ abolishes expression of glycogen metabolism and thermogenic genes. Chromatin profiling confirm that ERRs directly control the glycogen metabolic program in beige adipocytes. Together, our results identify a multilayered transcriptional axis that sustains glycogen metabolism during β-adrenergic activation in male mice.

Thermogenic adipocytes contribute to systemic energy expenditure by converting chemical energy into heat, a process predominantly regulated by sympathetic nervous system activation and β-adrenergic signaling[1,2]. Upon cold exposure or chronic adrenergic stimulation, white adipocytes can acquire beige characteristics, including mitochondrial biogenesis and expression of uncoupling protein 1 (UCP1)[3–5], enabling them to participate in non-shivering thermogenesis[6]. While this transformation is classically driven by the cAMP–PKA–CREB–PGC1α axis[7,8], emerging evidence suggests that additional metabolic pathways are integrated into the thermogenic program[9–12].

Our recent work identified a role for glycogen metabolism in the beigeing of subcutaneous adipocytes[13]. Chronic β-adrenergic stimulation promotes both glycogen accumulation and turnover in these

[1]Division of Endocrinology and Metabolism, Departments of Medicine and Pharmacology, University of California San Diego, San Diego, CA, USA. [2]Department of Cellular and Molecular Medicine, University of California, San Diego School of Medicine, San Diego, CA, USA. [3]Center for Epigenomics, University of California, San Diego School of Medicine, San Diego, CA, USA. [4]Gene Expression Laboratory, Salk Institute for Biological Studies, La Jolla, CA, USA. [5]Institute of Genomic Medicine, Moores Cancer Center, University of California, San Diego School of Medicine, San Diego, CA, USA. [6]Present address: Department of Genetics and Development, Columbia University Irving Medical Center, New York, NY, USA. [7]Present address: Department of Genetics and Development, Columbia University Irving Medical Center and New York Genome Center, New York, NY, USA. ✉e-mail: asaltiel@health.ucsd.edu

adipocytes, and this dynamic flux is necessary for beige adipogenesis by generating reactive oxygen species (ROS) necessary for activation of the p38 MAPK signaling pathway and subsequent expression of UCP1 and other thermogenic genes. Disruption of glycogen synthesis or turnover impairs this pathway, reducing thermogenic gene expression and cold-induced energy metabolism[13,14]. These findings indicate that glycogen is not merely a passive energy reservoir in adipose tissue, but an active metabolic node linking glucose utilization to thermogenic signaling.

Although these data implicated glycogen metabolism in white adipose tissue beiging, the transcriptional regulation of genes encoding the enzymes of glycogen metabolism in adipocytes remains poorly understood. In particular, how β3-adrenergic signaling induces this gene program and how this regulation is maintained during chronic thermogenic activation is uncertain. PGC1α (PPARGC1A) is a central transcriptional coactivator that regulates energy metabolism in adipocytes[15–18]. It is upregulated in response to cold exposure or β-adrenergic stimulation, promoting mitochondrial biogenesis and oxidative metabolism[2]. In brown and beige adipocytes, PGC1α drives the expression of thermogenic genes such as UCP1, enhancing heat production[15,19–21]. In white adipocytes, it also supports fatty acid oxidation and contributes to overall energy expenditure, playing a crucial role in maintaining metabolic health[22]. Estrogen-related receptors α and γ (ERRα and ERRγ) are orphan nuclear receptors that play crucial roles in regulating mitochondrial function and energy metabolism in adipose tissue[23,24]. Both ERRα and ERRγ act as key transcriptional regulators of oxidative metabolism by directly controlling the expression of genes involved in mitochondrial biogenesis[25,26], fatty acid oxidation[27,28], and thermogenesis[24,29–35]. In adipocytes, ERRα and ERRγ are activated through their interaction with PGC1α, forming a transcriptional complex that enhances the expression of metabolic and thermogenic genes, including Ucp1[24,36–40]. ERRα is particularly important for maintaining mitochondrial function and oxidative capacity in white and beige fat, while ERRγ is highly expressed in brown adipose tissue and contributes to its constitutively high oxidative and thermogenic activity[40,41]. Thus, the PGC1α – ERR axis is known to control oxidative and mitochondrial pathways[42], although its role in modulating glycogen metabolism in adipocytes is unknown.

Here, we dissect the regulation of a cassette of glycogen metabolism genes as a cell-autonomous component of the thermogenic program. We demonstrate that acute β-adrenergic stimulation of adipocytes selectively induces Gys2 and Ppp1r3c via PKA/CREB signaling, while the sustained expression of a subset of glycogen metabolic genes during chronic sympathetic activation requires PGC1α and ERRα/γ. Through chromatin accessibility profiling, loss-of-function genetics, and overexpression studies, we uncover a transcriptional circuit in which the PGC1α–ERR complex maintains glycogen turnover during prolonged adrenergic activation to support thermogenesis. These findings expand the known regulatory network of adipose thermogenesis and establish glycogen metabolism as a PGC1α–ERR–coordinated transcriptional module.

## Results

### β3 adrenergic receptor activation regulates glycogen metabolism genes in a cell autonomous fashion

To investigate the transcriptional regulation of glycogen metabolism during the browning of white adipose tissue, we examined RNA sequencing (RNA-seq) datasets that revealed a marked induction of genes involved in glycogen synthesis and degradation, including Gys1, Gys2, Pygl, Ppp1r3b, Ppp1r3c, and Agl, in inguinal white adipose tissue (iWAT) following 7 days of cold exposure or β3-adrenergic stimulation[43] (Fig. 1a). To determine the temporal dynamics of this response, we treated C57BL/6 J mice with the β3-adrenergic agonist CL-316,243 (CL) for 1, 3, 5, or 7 consecutive days. CL administration resulted in a progressive upregulation of Ucp1, consistent with the

emergence of beige adipocytes. Concomitantly, we observed a significant increase in the expression of muscle-enriched isoforms of glycogen synthase (Gys1), liver-enriched isoforms of glycogen synthase (Gys2) and liver glycogen phosphorylase (Pygl), as well as the glycogen debranching enzyme Agl (Fig. 1b, c). Notably, Gys2 was robustly induced at both the mRNA and protein levels within 24 hours of a single CL injection, whereas the upregulation of Pygl was delayed, becoming apparent only after 3 days of treatment (Fig. 1b, c, Supplementary Fig. 1a). Consistent with these molecular changes, PAS staining revealed progressive glycogen accumulation in parallel with beige adipocyte formation in iWAT of mice treated with CL for 1, 3, 5, or 7 consecutive days (Supplementary Fig. 1d).

To further define the early transcriptional response, we performed a high-resolution time-course analysis and confirmed that Gys2, as well as Ppp1r3c and Agl mRNAs are selectively and rapidly induced within the first 24 hours, alongside the thermogenic gene Ucp1(Fig. 1e). To test whether this response is cell autonomous, we treated differentiated adipocytes in vitro with CL for 6 h. This acute stimulation was sufficient to induce Gys2 and Ppp1r3c, but not other glycogen metabolic genes, suggesting selective and intrinsic regulation of glycogen synthesis in adipocytes (Supplementary Fig. 1b). Consistently, treatment with a cell permeable analog of cAMP for 6 h produced a similar induction of Gys2 and Ppp1r3c (Supplementary Fig. 1c). In contrast, primary mouse hepatocytes treated with cAMP or glucagon failed to induce these genes (Supplementary Fig. 1d, e). indicating that this regulation is adipocyte specific.

### CREB directly regulates Gys2 expression downstream of PKA signaling

To determine whether Gys2 and Ppp1r3c are direct transcriptional targets of protein kinase A (PKA) signaling, we treated primary adipocytes with the protein synthesis inhibitor cycloheximide prior to β3-adrenergic stimulation. Inhibition of de novo protein synthesis did not impair CL-induced expression of Gys2 and Ppp1r3c, indicating that both genes are primary transcriptional targets of PKA signaling and do not require intermediary protein synthesis (Fig. 2a). Pharmacological inhibition of PKA (H89), p38 MAPK (SB203580), or CREB (666-15) attenuated Gys2 mRNA induction in response to CL treatment in vitro (Fig. 2b), and CREB inhibition also suppressed CL-induced glycogen accumulation (Fig. 2c), suggesting a functional role for CREB in regulating glycogen metabolism.

To test directly the requirement for CREB in Gys2 regulation, we employed CRISPR–Cas9-mediated gene editing in primary pre-adipocytes differentiated in vitro. Using an adeno-associated virus (AAV) to deliver Creb-targeting sgRNAs into fully differentiated pre-adipocytes from Cas9 knock-in mice[44](Supplementary Fig. 2a), we achieved efficient knockdown (KD) of Creb mRNA and protein levels without affecting Atf2 mRNA expression (Fig. 2d, e). Consistent with this, CL-induced phosphorylation of CREB at serine[133] was markedly reduced (Fig. 2e, f). Loss of Creb abolished the induction of Gys2 by CL and reduced its basal expression (Fig. 2g). Glycogen accumulation was also diminished in Creb KD adipocytes in response to CL treatment (Fig. 2h). In contrast, Ppp1r3c expression was unaltered, suggesting that its regulation under PKA signaling is mediated by transcription factors other than CREB. Luciferase reporter assays showed that CREB significantly enhanced Gys2 promoter activity through a half CRE-binding site, and mutation of this site decreased the response (Supplementary Fig. 2b). These results indicate that CREB is essential for the transcriptional activation of Gys2 in adipocytes, but dispensable for the regulation of Ppp1r3c (Fig. 2g). Together, these findings identify Gys2 as a CREB-dependent gene downstream of β3-adrenergic signaling.

In humans, the expression levels of GYS2 in adipose tissue were negatively correlated with BMI (Supplementary Fig. 3b), GYS1 and GYS2

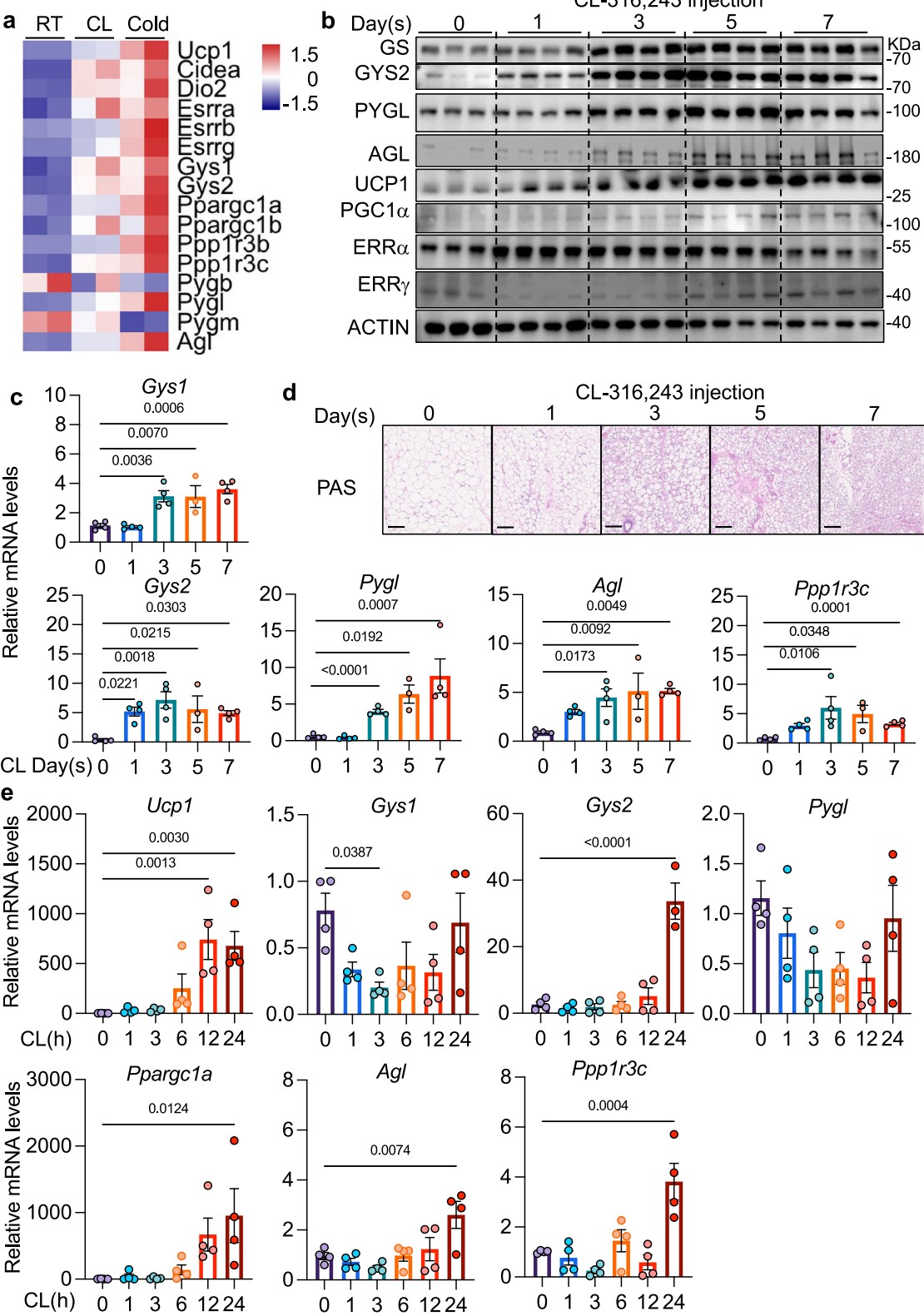

were negatively correlated with HOMA-IR, and Waist/hip ratio (Supplementary Fig. 3a, b). Higher expression of glycogen synthase genes is thus associated with lower body weight, higher insulin sensitivity, lower triglyceride, lower blood sugar and a non-obese phenotype (Supplementary Fig. 3a, b). These findings suggest a potential protective role of adipocyte glycogen synthase genes in maintaining metabolic health and preventing obesity[45–52].

## PGC1α sustains glycogen metabolism gene expression during chronic β3-adrenergic activation

Previous studies have shown that acute cold exposure or CL-316,243 treatment leads to desensitization of the β3-adrenergic receptor[53], prompting us to investigate how the expression of glycogen metabolism genes and glycogen turnover are sustained under chronic β3-adrenergic stimulation. We thus analyzed gene expression correlations

**Fig. 1 | β3-adrenergic receptor activation regulates glycogen metabolism genes in a cell-autonomous manner. a** Heat map showing transcriptional expression from RNA-seq analysis (adapted from GSE86338) of inguinal adipose tissue isolated from mice ($n = 2$) following chronic cold exposure (4 °C for 7 days) or β3-adrenergic agonist treatment (CL316,243, 1 mg/kg for 7 days), presented as z-scored FPKM values. **b, c** Protein and RNA expression levels of glycogen metabolism and thermogenic genes in iWAT of C57BL/6 J mice after the indicated days of CL treatment. $n = 4$ biological replicates for 0, 1, 3, 7-day CL injection, $n = 3$ biological replicates for WT for 5-day CL injection. Data are representative of two independent experiments with similar results. **d** Representative periodic acid–Schiff (PAS) staining of inguinal white adipose tissue (iWAT) sections from mice injected daily with CL-316,243 for the indicated days. Scale bars, 100 μm. **e** Gene expression levels of glycogen metabolism and thermogenic genes in iWAT of C57BL/6 J mice after the indicated hours of CL treatment. $n = 4$ biological replicates per treatment. Data (**b, c, e**) show mean ± s.e.m., *P* values were determined by two-sided Ordinary one-way followed by Dunnett's multiple comparisons test. Source data are provided as a Source Data file.

using a transcriptomic dataset from diversity outbred mice. This analysis revealed a strong positive correlation between glycogen metabolism genes and *Ppargc1a* (PGC1α) (Fig. 3a). Also, PPARGC1A expression shows a strong positive correlation with glycogen synthase and AGL in a human dataset[45–52] (Supplementary Fig. 4a), These findings suggest that PGC1α may coordinately regulate both glycogen synthesis (via GS) and breakdown (via PYGL and AGL) in adipose tissue, supporting metabolic flexibility under energy-demanding conditions.

To test the functional requirement of PGC1α in regulating glycogen metabolism in vivo, we generated adipocyte-specific PGC1α knockout mice (PGC1α-AKO) by crossing *Ppargc1a*-floxed mice with *AdipoQ-Cre* drivers. PGC1α expression was significantly decreased in PGC1α-AKO mice, without compensatory upregulation of *Ppargc1b* at either the RNA or protein level (Fig. 3b, Supplementary Fig 4b, c, 5a). In iWAT from PGC1α-AKO mice treated with CL for 7 days, the expression of glycogen metabolism genes, including *Gys1*, *Gys2*, *Ppp1r3c*, and *Agl* was markedly reduced, as was the canonical PGC1α downstream target *Ucp1* (Fig. 3b–d, Supplementary Fig 4b). Glycogen accumulation in iWAT was diminished in PGC1α-deficient mice following CL treatment (Fig. 3e). Furthermore, histology of iWAT from PGC1α-AKO mice revealed a marked reduction in CL-316,243-induced glycogen levels and UCP1 expression (Fig. 3f).

We also assessed glycogen metabolism in primary adipocytes differentiated in vitro from PGC1α fl/fl and PGC1α-AKO mice. Interestingly, there were no significant differences in expression of glycogen-related genes or *Ucp1* following 6 h of CL stimulation (Supplementary Fig. 5a), suggesting that PGC1α is dispensable for acute transcriptional responses, but required to sustain glycogen metabolic programs during chronic β3-adrenergic-induced browning of white adipocytes. Our previous study revealed that attenuation of glycogen synthesis or turnover reduced the activation of p38[13]. The activation of p38 by CL-316,243 treatment was not different between WT and PGC1α-AKO adipocytes (Supplementary Fig. 5b, c). The phosphorylation of HSL at Ser660/563 was also unchanged, indicating that loss of PGC1α does not impair adrenergic lipolysis (Supplementary Fig. 5b, c). Furthermore, beige adipocytes exhibited higher basal levels of glycogen metabolism genes compared to white adipocytes, and treatment with the selective PGC1α inhibitor SR-18292 significantly reduced these genes at both the mRNA and protein levels[54,55], reinforcing a role for PGC1α in maintaining their expression (Supplementary Fig. 5d–f).

To investigate whether PGC1α regulates chromatin accessibility at glycogen-related gene loci, we performed the assay for transposase-accessible chromatin with sequencing (ATAC-Seq) on iWAT from control and PGC1α-AKO mice following 7 days of CL treatment. Promoter regions of key glycogen metabolism genes exhibited reduced chromatin accessibility in the absence of PGC1α, suggesting impaired transcriptional competency (Fig. 3g). To test for direct PGC1α binding, we conducted chromatin immunoprecipitation (ChIP) assays and found that enrichment of PGC1α at the promoters of *Gys2*, *Ppp1r3c*, and *Agl* was abolished in PGC1α-AKO iWAT following CL treatment (Fig. 3h). These results collectively identify PGC1α as a critical regulator that maintains expression of glycogen metabolism genes during prolonged β3-adrenergic stimulation through direct chromatin engagement. Thus, PGC1α is a master regulator for glycogen metabolism genes in chronic β3 activation.

## PGC1α and ERRα/γ cooperatively maintain glycogen metabolism gene expression during chronic β3-adrenergic signaling

PGC1α is a well-established coactivator of multiple nuclear receptors in adipocytes, among which estrogen-related receptor α (ERRα) exhibits the highest affinity[23]. We first analyzed the RNA-seq dataset from brown adipose tissue derived from wild-type and *ERRα/γ* double knockout (*ERRα/γ* -AKO) mice treated with or without CL-316,243[39], and observed that the induction of glycogen metabolism genes was completely abolished in *ERRα/γ* -AKO mice (Supplementary Fig. 6a). Additionally, ChIP-seq analysis of ERRα and ERRγ revealed direct binding at the promoter regions of glycogen metabolism genes in brown fat[38,40] (Supplementary Fig. 6b).

To isolate the role of ERRα, we generated adipocyte-specific ERRα knockout (ERRα -AKO) mice by crossing *Esrra-flox/flox* mice with *AdipoQ-Cre*. Surprisingly, glycogen metabolism genes and thermogenic markers such as *Ucp1* remained unchanged in ERRα -AKO iWAT after CL treatment (Supplementary Fig. 7a–c), and glycogen accumulation was unaffected (Supplementary Fig. 7d). We observed a robust upregulation of *Esrrg* (ERRγ) expression in ERRα-AKO adipose tissue under both basal and CL-treated conditions (Supplementary Fig. 7a–c), consistent with prior reports that ERRγ not only increases in expression but can also functionally compensate for the loss of ERRα[29]. These findings suggested compensatory activation by ERRα homologs, particularly ERRγ, which shares 70–80% sequence similarity with ERRα in the DNA- and ligand-binding domains[32,56,57]. We also performed ERRγ ChIP–qPCR in ERRα-AKO mice, which revealed increased binding of ERRγ at promoter regions normally occupied by PGC1α and ERRα, indicating that enhanced ERRγ occupancy contributes to the redundant regulation of these target genes (Supplementary Fig. 8a).

To eliminate potential redundancy among ERR family members, we generated adipocyte-specific ERRα/β/γ triple knockout mice (ERRs-AKO). In these mice, chronic CL treatment failed to induce glycogen metabolism genes and thermogenic programs, including *Ucp1*, and glycogen content in iWAT was markedly reduced (Fig. 4a–e). Histological analysis confirmed a substantial reduction in both UCP1 protein and glycogen deposition in iWAT of ERRs-AKO mice following prolonged β3-adrenergic stimulation (Fig. 4f). To investigate whether ERRs directly modulate chromatin accessibility at glycogen-related gene loci, we performed ATAC-seq on iWAT from CL-treated control and ERRs-AKO mice. Chromatin accessibility at promoters of *Gys1*, *Gys2*, *Pygl*, *Ppp1r3c*, and *Agl* was significantly diminished in the absence of ERRs (Fig. 4g, Supplementary Fig. 6c). Furthermore, ChIP-seq analysis revealed that ERRα binding to these promoter regions was abolished in ERRs-AKO mice (Fig. 4g, h). Genome-wide profiling of ERRα binding revealed that occupancy is not limited to thermogenic and glycogen metabolic genes, but also includes loci associated with mitochondrial function and fatty acid metabolism, suggesting a broad regulatory role for ERRα in coordinating the transcriptional program of adipose browning (Fig. 4h). To functionally validate the role of each component, we overexpressed PGC1α, ERRα, or ERRγ individually in beige adipocytes. Overexpression of each factor led to the induction of *Gys1* and *Gys2* (Supplementary Fig. 8b). Together, these data demonstrate that during chronic β-adrenergic activation, PGC1α cooperates with ERRα and ERRγ to maintain glycogen metabolism gene

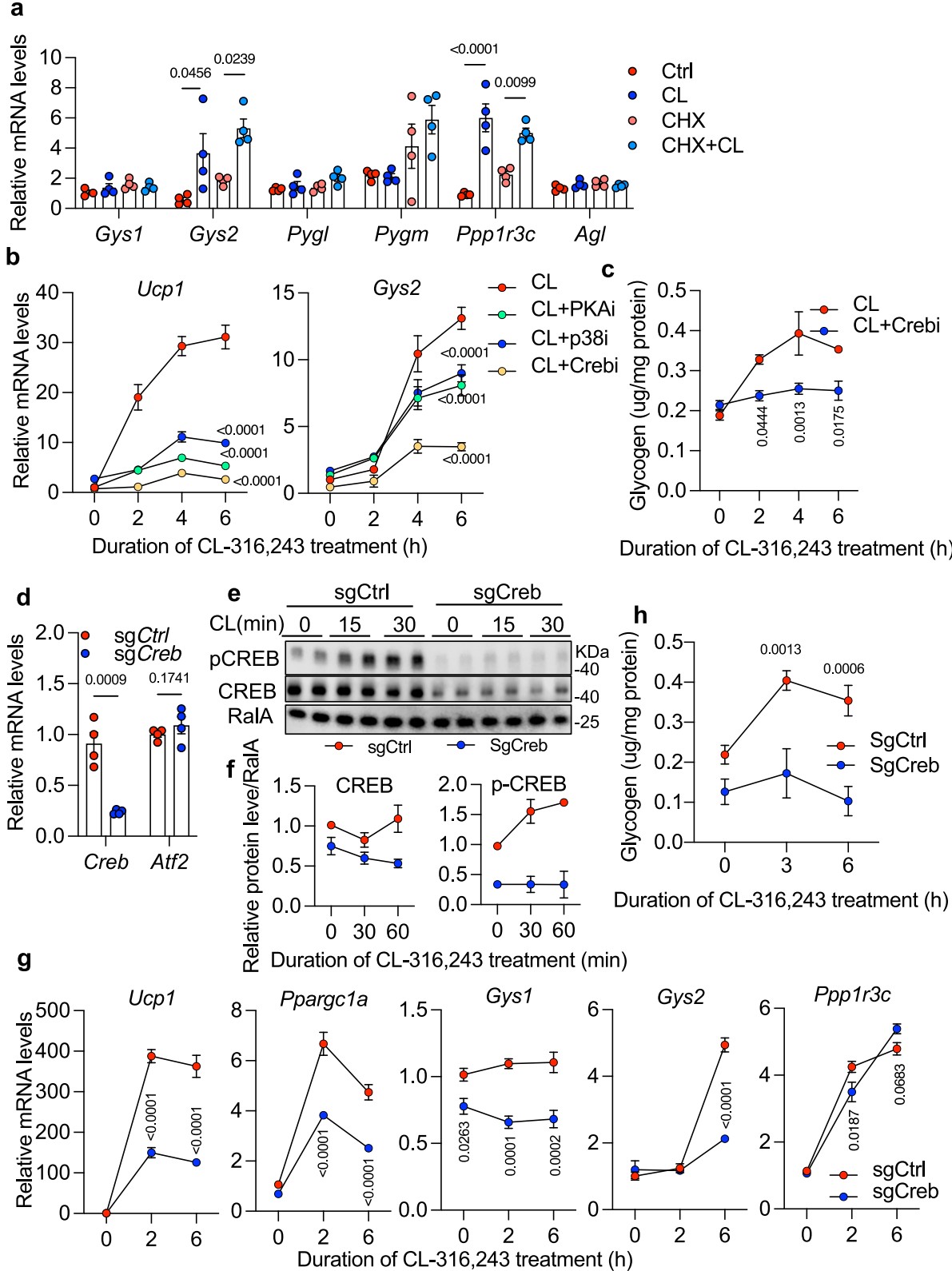

expression through direct engagement with chromatin, ensuring sustained glycogen turnover in thermogenic adipocytes (Supplementary Fig. 9).

## Discussion

Our findings identify a multilayered transcriptional program that connects β3-adrenergic signaling to glycogen metabolism in adipocytes. Acute β3-adrenergic receptor activation rapidly induces *Gys2* and *Ppp1r3c*, encoding key regulators of glycogen synthesis, via PKA signaling. With chronic stimulation, the induction of this gene program expands to include *Pygl* and *Agl*, which encode enzymes involved in subsequent hydrolysis and glycogen debranching, thus ensuring glycogen turnover. These changes are dependent on PGC1α and its interaction with ERRs, which bind directly to glycogen gene loci

**Fig. 2 | CREB directly regulates *Gys2* expression downstream of PKA signaling.** **a** Gene expression of glycogen metabolism genes in primary preadipocytes differentiated in vitro treated with CL-316,243 (3 h), with or without pretreatment with cycloheximide (CHX, 1 h). *n* = 4 biological replicates per treatment. **b** Gene expression of *Ucp1* and *Gys2* in response to CL-316,243 following primary preadipocytes differentiated in vitro with p38 inhibitor (SB203580, 10 µM), PKA inhibitor (H89, 10 µM), CREB inhibitor (666-15, 10 µM), or vehicle in wild-type preadipocytes. *n* = 4 biological replicates per treatment. **c** Glycogen levels in wild-type primary preadipocytes differentiated in vitro treated with CL-316,243 alone or in combination with 10 µM CREB inhibitor. *n* = 4 biological replicates per treatment. **d** *Creb* and *Atf2* mRNA expression in wild-type and CREB knockout (Creb KO) primary preadipocytes differentiated in vitro. *n* = 4 biological replicates per genotype per treatment. **e, f** Phosphorylation of CREB (**e**) and quantification of protein levels (**f**) in response to CL-316,243 in wild-type and Creb KO primary preadipocytes

differentiated in vitro. *n* = 2 biological replicates per genotype per treatment. Data are representative of two independent experiments with similar results. **g** Gene expression in wild-type and Creb KO primary preadipocytes differentiated in vitro following indicated durations of CL-316,243 treatment. *n* = 4 biological replicates per genotype per treatment. **h** Glycogen levels in wild-type and *Creb* KO primary preadipocytes differentiated in vitro following indicated durations of CL-316,243 treatment. *n* = 4 biological replicates per genotype per treatment. Data (**a**) show mean ± s.e.m., *P*-values were determined by two-sided Ordinary one-way with Tukey's multiple comparisons test. Data (**d**) show mean ± s.e.m., *P*-values were determined by two-sided Unpaired t tests with false discovery rate control using the Benjamini, Krieger and Yekutieli method. Data (**b, c, g, h**) show mean ± s.e.m., *P*-values were determined by two-sided Two-way ANOVA followed by Tukey's multiple comparisons test for (**b**), Šídák's multiple comparisons test for (**c, g, h**). Source data are provided as a Source Data file.

and maintain their transcriptional activity. Adipocyte-specific deletion of ERR isoforms abrogates this gene expression pattern, establishing the PGC1α–ERR axis as a central regulator of adrenergically-driven glycogen remodeling (Supplementary Fig. 9). Together, these data position glycogen metabolism as a transcriptionally regulated effector branch of thermogenic signaling in adipocytes.

While our study defines a key role for the PGC1α–ERR module in maintaining glycogen gene expression during chronic adrenergic stimulation, it is unlikely to be the sole mechanism. Glycogen metabolism genes are subject to multiple layers of transcriptional regulation, including PKA-dependent induction, chromatin remodeling, and potential inputs from other nuclear receptors or co-regulators. Our findings highlight one transcriptional axis that becomes dominant under prolonged thermogenic stimulation, but likely represents a subset of a broader regulatory network. Similar to the multilayered control of thermogenic programs in adipocytes[9,20], the transcriptional regulation of glycogen metabolism is complex and intricately organized. Dissecting how different transcription factors and epigenetic states converge to fine-tune glycogen gene expression in response to varying metabolic cues remains an important avenue for future research. Our findings establish CREB as a direct regulator of *Gys2* transcription, with sgRNA specificity confirmed by the absence of effects on *Atf2* expression. However, given that p38 inhibition attenuates *Gys2* induction, we cannot exclude a contributory role of ATF2, and future work will be needed to define how CREB and ATF2 cooperate in regulating glycogen metabolism.

The integration of carbohydrate metabolism with mitochondrial gene programs by PGC1α and ERRs suggests a coordinated strategy to enable metabolic flexibility during thermogenesis[21,22,58]. While PGC1α is best known for its role in oxidative metabolism[42,59], our results extend its transcriptional reach to cytosolic glucose-handling enzymes. This dual regulatory function may allow adipocytes to efficiently match substrate availability with mitochondrial demand. Importantly, this regulatory pathway represents a mechanism to ensure that only adipocytes with sufficient fuel reserves to undergo thermogenesis and thus survive the potential toxic effects of UCP1 expression are able to respond to beiging signals.

Our data also raise the question of how chronic β3-adrenergic stimulation leads to selective recruitment of the PGC1α–ERR module at glycogen-related loci. Chromatin accessibility at these promoters is largely dependent on this axis, yet the mechanisms that guide PGC1α and ERRs to specific genomic targets under sustained adrenergic tone remain unclear. Identifying upstream signaling intermediates or chromatin remodelers involved in this selective enhancer engagement could offer insight into how adipocytes establish and maintain thermogenic competency[12].

In line with prior literature[15,16,18] PGC1α is a well-established coactivator for *Ucp1* transcription. Interestingly, in our adipose-specific PGC1α-AKO model, *Ucp1* expression was only modestly reduced. This raises the possibility that PGC1β may compensate for the complete loss

of PGC1α. Indeed, both mRNA and protein analyses confirmed that PGC1β levels are maintained in adipose tissue lacking PGC1α, suggesting that functional redundancy within the PGC1 family may buffer against the loss of a single isoform. While our study primarily focused on PGC1α, these observations highlight the importance of considering cooperative or compensatory interactions between PGC1α and PGC1β in sustaining adipocyte thermogenic and glycogen metabolic programs.

A key limitation of our study is that the compensatory upregulation of ERRγ observed in ERRα-AKO mice has not yet been evaluated in human adipocytes. While isoform redundancy among ERRs is well established in murine systems, the extent to which similar mechanisms operate in human adipose tissue remains unknown. Direct genetic ablation strategies that enable rigorous testing in mice are not readily applicable to primary human adipocytes, making this question difficult to address within the scope of the present study. Nonetheless, given the translational importance of ERR signaling in metabolic regulation, future studies using human adipocyte models, such as CRISPR-based perturbations or organoid systems, will be valuable to determine whether isoform compensation represents a conserved regulatory strategy across species.

In addition to this transcriptional control, glycogen metabolism is also subject to rapid and reversible regulation at the post-translational level. Glycogen synthase activity is inhibited by phosphorylation through kinases such as GSK3, AMPK, and PKA, and reactivated by PP1-mediated dephosphorylation[60–62]. Conversely, glycogen phosphorylase is activated by phosphorylation via phosphorylase kinase, while PTG recruits PP1 to glycogen particles to promote dephosphorylation of both synthase and phosphorylase[63]. These well-established mechanisms ensure that glycogen turnover can be acutely adjusted to hormonal and nutrient inputs, complementing the transcriptional program we describe here.

Finally, these findings expand the potential utility of targeting the PGC1α–ERR pathway beyond mitochondrial regulation alone. Pharmacologic or genetic strategies to enhance ERR activity have shown promise in improving oxidative metabolism[64–66], and our work now suggests that modulating glycogen turnover could represent an additional therapeutic angle. Whether enhancing glycogen flux independently of mitochondrial biogenesis can drive adipocyte thermogenesis or systemic energy expenditure remains a compelling direction for future work.

## Methods

### Animals

C57BL/6 J (strain no. 000664), *PGC1α* fl/fl (strain no. 009666), and *Adipoq*-Cre (strain no. 028020), Rosa26-Cas9 knockin on B6J (strain no. 026179) mice were obtained from The Jackson Laboratory. *PGC1α* fl/fl mice were crossed with *Adipoq*-Cre mice to generate adipocyte-specific *PGC1α* knockout mice. *Esrra* (ERRα) single knockout (ERRα-AKO) and *Esrra/Esrrb/Esrrg* triple knockout (ERRs-AKO) mice were

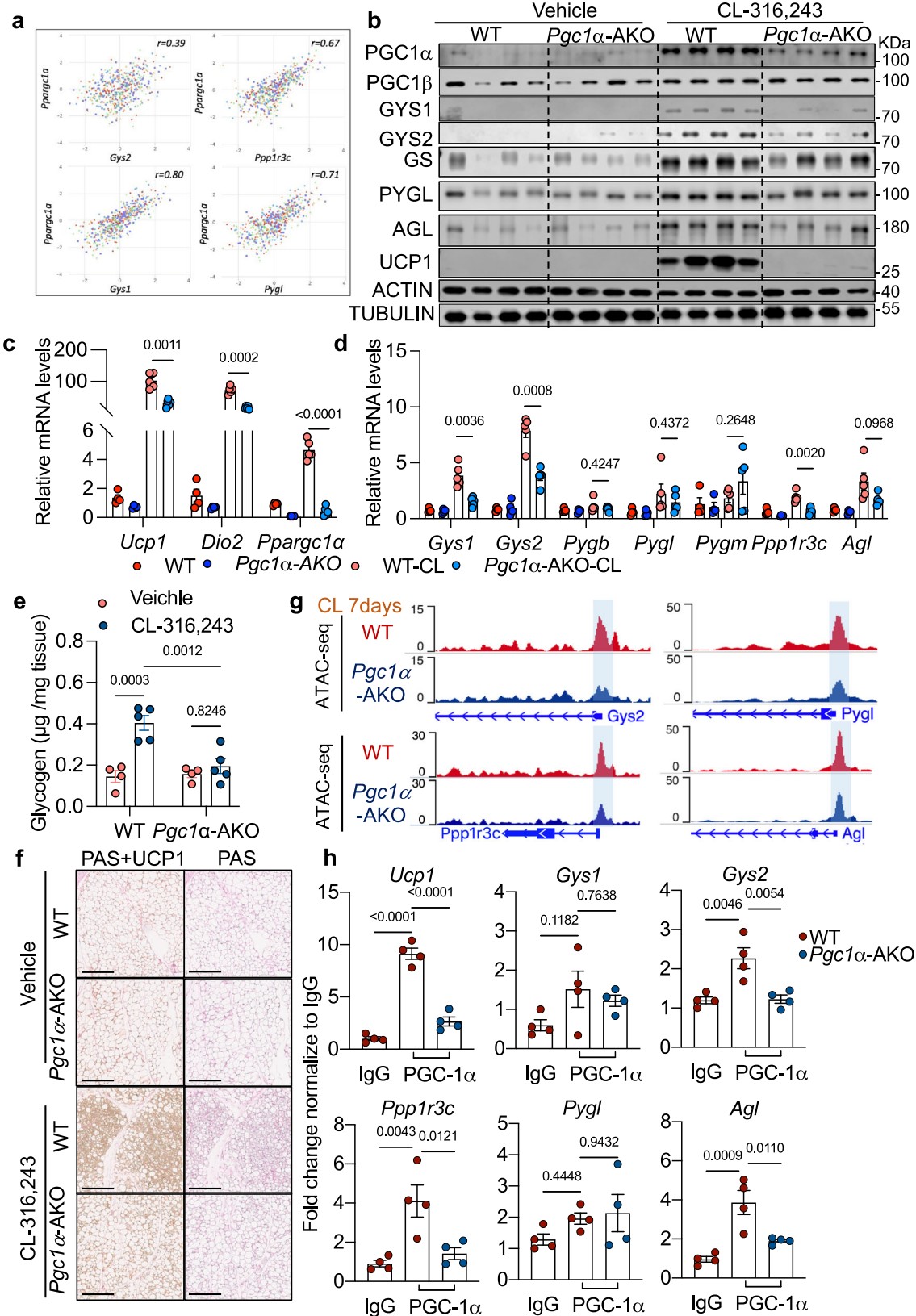

provided by Dr. Ronald Evans; all mice were maintained on a C57BL/6J background[67]. Male mice aged 8–12 weeks were maintained on standard chow (7912, Teklad) under specific pathogen-free (SPF) conditions with 12 h:12 h light–dark cycles and ad libitum access to food and water. For in vivo studies, CL-316,243 (1 mg/kg) was administered intraperitoneally. All animal procedures were approved by the

Institutional Animal Care and Use Committee (IACUC) at the University of California, San Diego.

## Cell culture

**Preadipocyte Isolation.** Preadipocytes were isolated from the inguinal white adipose tissue (WAT) of 8- to 10-week-old mice. Inguinal fat was

**Fig. 3 | PGC1α sustains glycogen metabolism gene expression during chronic β3-adrenergic activation. a** Gene expression correlations in adipose tissue from diversity outbred strains of high-fat diet (HFD)-fed mice. **b** Protein expression in inguinal white adipose tissue (iWAT) of wild-type (WT) and PGC1a adipocyte-specific knockout (PGC1α-AKO) mice treated with vehicle or CL-316,243 for 7 days. *n* = 4 mice per genotype per treatment. Data are representative of two independent experiments with similar results. **c**, **d** Gene expression in iWAT of mice treated as in (**b**). *n* = 4 mice for non-CL-treated WT and PGC1α-AKO groups, *n* = 5 for 7-day CL treated WT and PGC1α-AKO groups. **e** Glycogen levels in iWAT from WT and PGC1α-AKO mice treated as in (**b**). *n* = 4 mice for non-CL-treated WT and PGC1α-

AKO groups, *n* = 5 mice for 7-day CL treated WT and PGC1α-AKO groups. **f** Glycogen accumulation and UCP1 expression analyzed by PAS staining and immunohistochemistry, respectively. Scale bars, 200 μm. **g** ATAC-seq analysis of gene loci in iWAT from WT and PGC1α-AKO mice treated as in (**b**). **h** ChIP−qPCR assay using an antibody against PGC1a in iWAT from WT and PGC1α-AKO mice treated as in (**b**). *n* = 4 mice per genotype per treatment. Data (**c**, **d**) show mean ± s.e.m., *P*-values were determined by a two-sided unpaired *t*-test with Welch's correction. Data (**e**, **h**) shows mean ± s.e.m., *P*-values were determined by two-sided Ordinary one-way ANOVA followed by Tukey's multiple comparisons test. Source data are provided as a Source Data file.

finely minced and digested in 10 mL of serum-free DMEM/F12 containing 1 mg/mL collagenase (C6885, Sigma) for 30 min at 37 °C with gentle agitation. Digestion was stopped by adding DMEM/F12 containing 10% FBS. The tissue suspension was filtered through a 100-μm strainer and centrifuged at 750 × g for 10 min at room temperature. The supernatant was discarded, and the pellet was resuspended in DMEM/F12 with 10% FBS, then filtered through a 40 μm strainer and centrifuged again at 750 × g for 10 min. The final pellet was resuspended in DMEM/F12 containing 10% FBS and plated onto 10 cm tissue culture dishes in DMEM/F12 with 15% FBS. After 2 days, the medium was removed, and non-adherent cells were washed away with PBS. Fresh culture medium containing 10% FBS was added. Once the cells reached ~70% confluence, they were replated for experiments and maintained in DMEM/F12 with 10% FBS, with medium changes every 2–3 days.

Differentiation was initiated 1–2 days after cultures reached full confluence by adding 500 μM 3-isobutyl-1-methylxanthine, 250 nM dexamethasone, 1 μg/mL insulin, and 1 μM troglitazone for 4 days. The medium was then changed to DMEM/F12 with 10% FBS containing 1 μg/mL insulin for 3 days, followed by maintenance in DMEM/F12 with 10% FBS. Cells were used for experiments on day 8 or 9 after the initiation of differentiation. AAV8-sgControl and AAV8-sgCreb, both carrying Cre, were introduced into differentiated adipocytes derived from Rosa26-Cas9 knock-in B6J mice. Protein and mRNA levels were measured 48–72 h after transduction. All treatments were conducted in serum-free DMEM, with 2% FFA-free BSA as indicated.

For beige adipocyte differentiation, stromal vascular fraction (SVF) cells were cultured in DMEM containing 10% FBS, 5 μg/mL insulin, 1 nM T3, 1 μM rosiglitazone, 0.5 mM isobutylmethylxanthine, 250 nM indomethacin, and 250 nM dexamethasone. After 48 hours, cells were cultured in medium containing 10% FBS, 5 μg/mL insulin, 1 nM T3, and 1 μM rosiglitazone for another 5–7 days. Only cultures in which >90% of cells displayed mature adipocyte morphology were used. All media were supplemented with 10 U/mL penicillin and 10 U/mL streptomycin.

**Primary Hepatocyte Isolation**
Mice were anesthetized and subjected to liver perfusion with 15 ml of calcium-free HEPES−phosphate buffer (pH 7.4), followed by 25 ml of HEPES−phosphate buffer (pH 7.4) containing 40 μg/ml Liberase TM (Roche). A final perfusion step was performed using 25 ml of calcium-free HEPES−phosphate buffer (pH 7.4). The liver was then excised, and primary hepatocytes were released by mechanical dissociation in 30 ml of cold HEPES−phosphate buffer (pH 7.4). The resulting cell suspension was filtered through a 70-μm nylon mesh (Corning) and centrifuged at 50 × g for 5 min. The supernatant was discarded, and the pellet was resuspended in 50 ml of HEPES−phosphate buffer (pH 7.4) containing 36% Percoll, followed by centrifugation at 100 × g for 10 minutes. The enriched hepatocyte pellet was resuspended in 10 ml of pre-warmed William's E medium (Life Technologies) supplemented with 10% FBS, 10 mM HEPES, 2 mM L-glutamine, 8 mg/L gentamicin, SPA, 1 μM dexamethasone, 4 μg/ml insulin, and 1 mM glucose. Cells were seeded onto collagen-coated plates at a density of 4 × 10^5 cells/ml. Four hours post-plating, the medium was replaced to remove non-

adherent and dead cells. For glucagon and cAMP treatments, hepatocytes were pre-incubated for 1 hour in FBS-free William's E medium supplemented with 10 mM HEPES, 2 mM L-glutamine, 8 mg/L gentamicin, SPA, 1 μM dexamethasone, and 1 mM glucose. All treatments were conducted in this FBS-free medium.

**Glycogen extraction**
Glycogen extraction was performed as previously described with slight modifications[13,68]. iWAT samples were defatted by incubation in 1 mL of a methanol: chloroform solution (1:2 ratio) for 30 min at room temperature. Samples were centrifuged at 6000 × g for 5 min, the solution was aspirated, and the process was repeated once more. After the second centrifugation, samples were allowed to dry completely before proceeding. This defatting step was not applied to cells, liver, or BAT tissues.

Samples were then boiled for 30 min in 500 μL of 30% KOH solution. After cooling, 100 μL of 1 M $Na_2SO_4$ was added, followed by 1.2 mL of pure ethanol. Samples were boiled for 5 min and centrifuged at 16,000 × g for 5 min. Pellets were washed twice by resuspension in 500 μL of double-distilled water, followed by the addition of 1 mL of pure ethanol and a brief boil. After the final wash, pellets were completely air-dried and resuspended in 150 μL of 50 mM sodium acetate buffer (pH 4.8) containing 0.3 mg/mL amyloglucosidase and incubated at 37 °C overnight.

Glycogen content was quantified using the Autokit Glucose assay (997-03001, Fujifilm), with comparison to a standard curve.

**RNA extraction, RT−PCR and qPCR**
For RNA isolations from in vivo experiments, following dissection, tissues were immediately snap frozen in liquid nitrogen and stored at −80 °C until processing. Tissues were directly homogenized in TRIzol (15596018, Life Technologies) according to the manufacturer's instructions. One micrograms of RNA were used for cDNA synthesis using HiScript III RT SuperMix for qPCR (+ gDNA wiper) (R323-01, Vazyme). Quantitative PCR was performed in quadruplicate using the Taq Pro Universal SYBR qPCR Master Mix (Q712-03, Vazyme). The Applied Biosystems QS5 real-time PCR System was used with the standard curve settings.

**Protein Extraction and Immunoblotting**
Tissues or cells were lysed or homogenized in denature buffer (50 mM Tris-HCl, pH7.5, 0.5 M EDTA, 1%SDS, 1 mM DTT) supplemented with freshly added Halt Protease and Phosphatase Inhibitor Cocktail (Thermo Fisher). Lysates were rotated at 4 °C for 30 min, and then centrifuged at 17,000 × g for 15–20 minutes at 4 °C. The cleared supernatants were collected, and protein concentrations were determined using a BCA Protein Assay Kit. Proteins were separated by electrophoresis on Tris-Glycine gels (Novex, Invitrogen) and transferred to PVDF membranes. The blots were blocked with 5% BSA in TBST (0.1% Tween-20 in TBS) at room temperature for 1 h and incubated with primary antibodies diluted in blocking buffer overnight at 4 °C. Membranes were incubated with the appropriate secondary antibodies conjugated with horseradish peroxidase for 2 h at room temperature SuperFemto ECL Chemiluminescence Kit (E423-02,

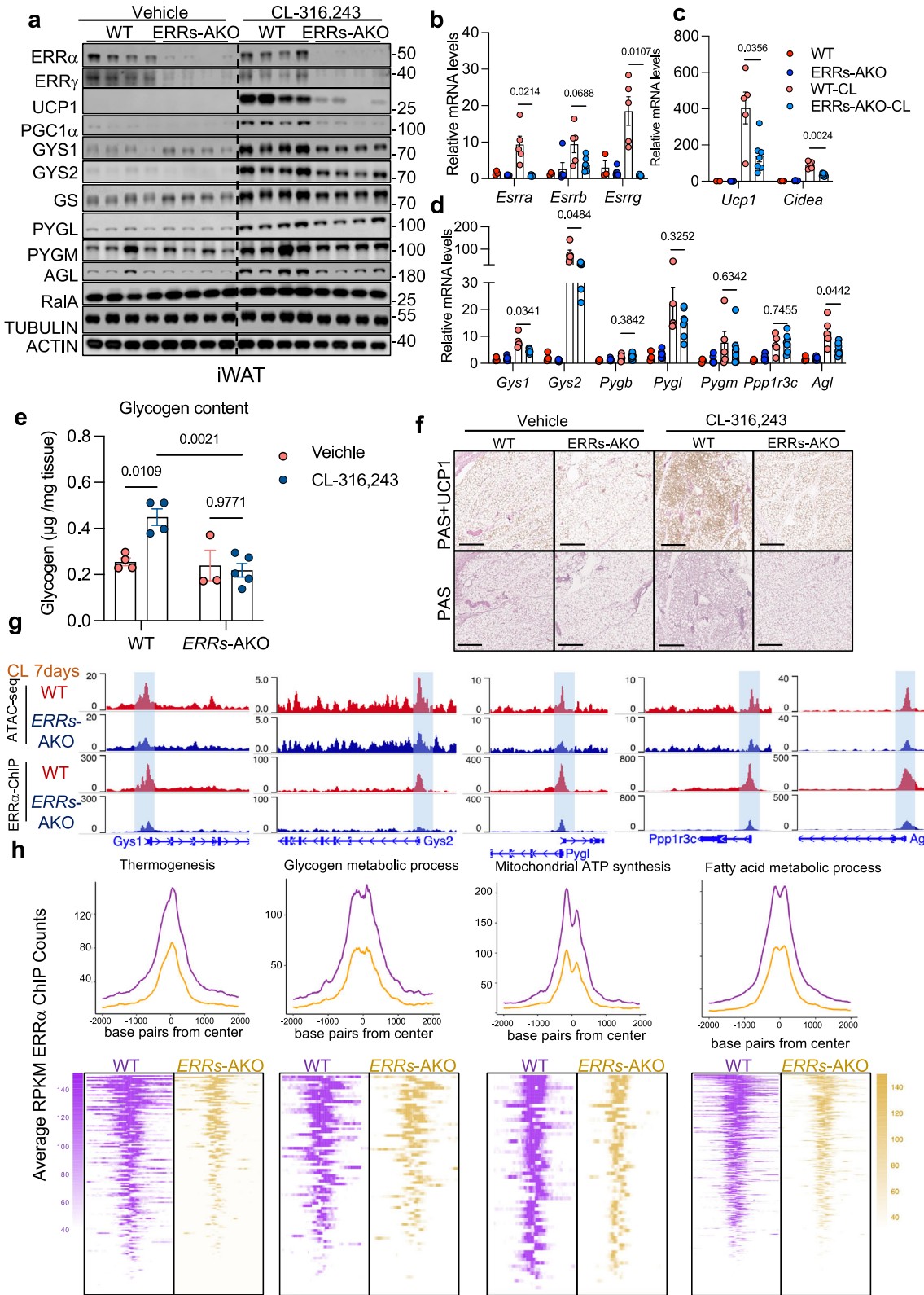

Vazyme) and Li-Cor detection system (Fisher Scientific) were used for Western blot detection.

## Antibodies

GYS1 (10566-1-AP), GYS2 (22371-1-AP), PYGL (15851-1-AP), PYGM (19716-1-AP), AGL (16582-1-AP), DIO2 (26513-1-AP), GFP (50430-2-AP) were obtained from Proteintech. pHSL-563 (4139S), pHSL-660 (45804S), HSL (4107S), p38 (9218S), p-p38 (9211), Tubulin (3873S), Actin (4970S),

ERRα (13826S) were obtained from Cell Signaling Technology. PGC1α (AB3242, Sigma). UCP1 (AB209483) and ERRγ (Ab219547) were obtained from Abcam. ERRγ antibody was generated by Dr. Ronald Evans laboratory.

## Histology

Tissues were collected and fixed in 10 % formalin for 48 h and then washed and stored in 70% ethanol until processing. All the following

**Fig. 4 | PGC1α and ERRα/γ cooperatively maintain glycogen metabolism gene expression during chronic β3-adrenergic signaling. a** Immunoblot analysis of iWAT from wild-type (WT) and adipocyte-specific ERRs knockout (ERRs-AKO) mice treated with vehicle or CL-316,243 for 7 days. $n = 4$ mice per genotype per treatment. Data are representative of three independent experiments with similar results. **b–d** Relative mRNA expression of glycogen metabolism and thermogenic genes in iWAT of mice treated as in (**a**). $n = 3$ biological replicates for WT, $n = 5$ biological replicates for ERRs-AKO, $n = 5$ biological replicates for WT-CL, $n = 7$ biological replicates for ERRs-AKO-CL. **e** Glycogen content in iWAT of WT and ERRs-AKO mice following 7 days of CL-316,243 or vehicle treatment. $n = 4$ biological replicates for non-CL treated and 7-day CL treated WT groups, $n = 3$ biological

replicates for non- CL treated ERRs-AKO group, and $n = 5$ biological replicates for 7-day CL treated ERRs-AKO groups. **f** Glycogen accumulation and UCP1 expression in iWAT assessed by PAS staining and immunohistochemistry, respectively. Scale bars, 200 µm. **g** Chromatin accessibility (ATAC-seq) and ERRα occupancy (ChIP–seq) at representative loci in iWAT of WT and ERRs-AKO mice treated as in **a**. **h**, Heatmap showing ChIP–seq peak tag density for ERRα across gene loci grouped by functional categories. Peak intensity is centered on the binding summit ($\pm 2$ kb) and ranked by signal strength. Data (**b–d**) show mean $\pm$ s.e.m., P-values were determined by a two-sided unpaired t-test with Welch's correction. Data (**e**) was determined by two-sided Two-way ANOVA followed by Tukey's multiple comparisons test. Source data are provided as a Source Data file.

steps such as paraffin embedding, sectioning, hematoxylin and eosin staining and immunohistochemistry were done at the UCSD Tissue Technology Core. For Periodic Acid–Schiff (PAS) Staining, paraffin or frozen sections (4–5 µm) were baked at 60 °C for 1 h, deparaffinized in xylene (two changes, 3 min each), and rehydrated through graded ethanol (100%, 95%, 70%) into distilled water. Sections were incubated in 0.5% periodic acid for 5 min, rinsed in distilled water, and stained with Schiff's reagent at room temperature for 15 min until a deep magenta color developed. After washing in running tap water for 5 min, sections were counterstained with hematoxylin for 1 min, clarified, and blued. Slides were dehydrated through graded ethanol, cleared in xylene, and mounted with a coverslip. PAS staining highlights glycogen, basement membranes, and mucosubstances in bright magenta. For immunohistochemistry (IHC), paraffin-embedded sections were baked at 60 °C for 1 h, deparaffinized in xylene, and rehydrated through graded ethanol to distilled water. Antigen retrieval was performed using citrate-based antigen unmasking solution (pH 6.0, Vector) at 95 °C for 30 min. Staining was performed on an Intellipath Automated IHC Stainer (Biocare). Endogenous peroxidase activity was blocked with Bloxall (Vector) for 10 min, followed by washing in TBST and blocking with Blotto (Thermo) for 10 min. Sections were incubated with primary antibody for 1 h, washed, and incubated with an anti-species HRP polymer secondary reagent for 30 min. DAB chromogen was applied for 5 min, and slides were rinsed in water, counterstained with Mayer's hematoxylin for 5 min, dehydrated, cleared, and mounted with a xylene-based medium.

## Chromatin Immunoprecipitation (ChIP)

ChIP was performed using the SimpleChIP® Enzymatic Chromatin IP Kit (#9003, Cell Signaling Technology) with slight modifications. Briefly, immediately after euthanasia, inguinal white adipose tissue (iWAT) depots from ten mice treated with CL316,243 for 7 days were snap-frozen. Frozen tissues were later thawed and finely minced in 10 ml of cross-linking buffer (1× PBS with 1% formaldehyde), followed by incubation on a rocker for 20 min at room temperature. Cross-linking was quenched by adding 0.5 ml of 2.5 M glycine and incubating for 5 min with mixing. Tissues were then washed three times with ice-cold 1× PBS. Cross-linked tissues were resuspended in ChIP dilution buffer containing protease inhibitors (cOmplete™, Roche) and kept on ice. Nuclei were pelleted by centrifugation at 2000 × g for 5 min at 4 °C. The supernatant was discarded, and the pellet was resuspended in 1 ml of ice-cold 1× Buffer B with DTT per IP preparation. After another centrifugation step, the pellet was resuspended in 100 µl of 1× Buffer B + DTT and transferred to a 1.5 ml microcentrifuge tube. For chromatin digestion, 0.5 µl of Micrococcal Nuclease was added per IP, mixed by inversion, and incubated at 37 °C for 20 min with gentle mixing every 3–5 min to achieve DNA fragments of ~150–900 bp. The reaction was stopped by adding 10 µl of 0.5 M EDTA and cooling on ice. Nuclei were pelleted at 16,000 × g for 1 min at 4 °C. The pellet was then resuspended in 100 µl of 1× ChIP buffer with protease inhibitors and incubated on ice for 10 minutes. Lysates (up to 500 µl) were sonicated with short pulses, placing the samples on wet ice for 30 s between pulses to disrupt nuclear membranes. Lysates were clarified

by centrifugation at 9,400 × g for 10 minutes at 4 °C. For each ChIP, 500 µl of diluted chromatin was incubated with specific antibodies at 4 °C for 4 hours to overnight with rotation. Protein G magnetic beads (30 µl per IP) were added and incubated for an additional 2 hours at 4 °C with rotation. Beads were pelleted using a magnetic rack, and the supernatant was carefully removed. Beads were washed three times with low-salt wash buffer (1 ml, 5 min each at 4 °C with rotation), followed by one high-salt wash under the same conditions. DNA was eluted and purified according to the kit protocol. Sequencing libraries were generating using the KAPA Hyper Prep Kit (KAPA Biosystems, Wilmington, MA, USA) following manufacturer's instructions using 13 cycles of amplification. The quality of the library was assessed using High Sensitivity D1000 kit on a 4200 TapeStation instrument (Agilent Technologies, Santa Clara, CA, USA). Sequencing was performed using the NovaSeq X Plus Sequencing System (Illumina, San Diego, CA, USA), generating 100 bp paired-end reads to obtain ~40 M reads per sample.

Sequencing reads were aligned to the mouse reference genome (mm10) using Bowtie2 (v2.4.5) with local alignment mode and the --no-unal option. The resulting SAM files were converted to BAM format using samtools view (v1.15) and subsequently sorted with samtools sort. Duplicated reads were removed using Picard (v2.26.10)'s Mark-Duplicates function.

## ChIP-seq promoter enrichment analysis

To compare enrichment of ChIP-seq signal at gene promoters of interest, first RPKM normalized bigwig files were generated using bamCoverage from deeptools version 3.5.2. The ComputeMatrix reference-point function from deeptools version 3.5.2 was used to compute ChIP-seq signal in 10 bp bins −2000 bp to +2000 bp relative to transcription start sites annotated in mm10 Gencode Release 23. To visualize, heat maps and average profile plots were generated.

## ATAC-seq

Permeabilized nuclei were obtained by resuspending grinding tissue in 1 mL Nuclear Permeabilization Buffer (0.2% IGEPAL-CA630 (I8896, Sigma), 1 mM DTT (D9779, Sigma), Protease inhibitor (05056489001, Roche), 5% BSA (A7906, Sigma) in PBS (10010-23, Thermo Fisher Scientific)), and incubating for 10 min on a rotator at 4 °C. Nuclei were then pelleted by centrifugation for 5 min at 500 x g at 4 °C. The pellet was resuspended in 25 µL ice-cold Tagmentation Buffer [33 mM Tris-acetate (pH = 7.8) (BP-152, Thermo Fisher Scientific), 66 mM K-acetate (P5708, Sigma), 11 mM Mg-acetate (M2545, Sigma), 16 % DMF (DX1730, EMD Millipore) in Molecular biology water (46000-CM, Corning)]. An aliquot was then taken and counted by hemocytometer to determine nuclei concentration. Approximately 50,000 nuclei were resuspended in 20 µL ice-cold Tagmentation Buffer, and incubated with 1 µL Tagmentation enzyme (FC-121-1030; Illumina) at 37 °C for 60 min with shaking 500 rpm. The tagmented DNA was purified using MinElute PCR purification kit (28004, Qiagen). The libraries were amplified using NEBNext High-Fidelity 2X PCR Master Mix (M0541, NEB) with primer extension at 72 °C for 5 min, denaturation at 98 °C for 30 s, followed by 8 cycles of denaturation at 98 °C for 10 s, annealing at 63 °C for 30 s and extension at 72 °C for 60 s. Amplified libraries were

then purified using MinElute PCR purification kit (28004, Qiagen), and two size selection steps were performed using SPRIselect bead (B23317, Beckman Coulter) at 0.55X and 1.5X bead-to-sample volume rations, respectively. Libraries were sequenced to the dept of 80 M reads.

## Statistical analysis

Statistical analyses were conducted using GraphPad Prism 9. Data are presented as mean ± s.e.m. Comparisons between two groups were performed using unpaired Student's t-tests. For comparisons involving more than two groups, one-way or two-way ANOVA was used, as specified in figure legends. *P*-values are indicated in the figures. No data were excluded from analysis unless due to technical issues or failure of control conditions. During tissue processing for RNA extraction, RT–PCR, and lysis, sample blinding was achieved by random allocation without treatment knowledge. Investigators were otherwise not blinded. Variance was not formally estimated between groups, and sample sizes were not predetermined, but instead based on experimental feasibility and animal or cell availability. When applicable, mice were randomly assigned to treatment or control groups, with treatments randomized across cages. All in vivo experiments were replicated in at least two independent cohorts. In vitro experiments were not randomized.

## Reporting summary

Further information on research design is available in the Nature Portfolio Reporting Summary linked to this article.

## Data availability

The ChIP-seq and ATAC-seq data generated in this study have been deposited in the NCBI SRA database under accession code PRJNA1260692. All data supporting the findings of this study are available within the paper and its Supplementary Information. Source data are provided as a Source Data file. qPCR primer sequences are provided in Supplementary Table 1. Source data are provided with this paper.

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

## Acknowledgements

We thank members of the Saltiel lab and collaborators for helpful discussions. H.F. was supported by American Diabetes Association postdoc fellowship 315903-00001. B.Z. was supported by American Heart Association postdoc fellowship 24POST1187840. Tissue Technology Shared Resource is supported by a National Cancer Institute Cancer Center Support Grant CCSG P30CA23100. This publication includes data generated at the UC San Diego IGM Genomics Center with funding from a National Institutes of Health SIG grant S10 OD026929. R.M.E. was supported by the NOMIS Foundation, and NIH R01DK057978. This work was supported by a grant from the Larry L. Hillblom Foundation to R.M.E. and A.R.S., US National Institute of Health (NIH/NIDDK) grants P30DK063491, P30DK120515, R01DK122804, R01DK124496, R01DK125820 and R01DK128796 to A.R.S.

## Author contributions

A.R.S. and H.F. conceived the project. H.F. designed and performed experiments, interpreted data, prepared figures, and wrote the manuscript. Y.W., J.G., D.T., B.V., B.Z., X.M., J.Z., T.D., and Y.R. performed experiments. W.F., M.D., and R.M.E. generated the ERR-related mouse models. S.L., N.R.Z., and B.R. conducted the bioinformatic analysis of ChIP-seq and ATAC-seq data and wrote the corresponding methods.

A.R.S. directed the project, designed experiments, interpreted data, and wrote the manuscript.

## Competing interests

The authors declare no competing interests.
