## [Transparent Peer Review file · Nature Communications]

Multi-layered Transcriptional Control of Glycogen Metabolism Coordinates Thermogenic Remodeling of White Adipocytes

Corresponding Author: Dr Alan Saltiel

Version 0:

Reviewer comments:

Reviewer #1

(Remarks to the Author)

This manuscript investigates the regulation of glycogen in adipocytes during the process of β 3AR-induced beiging. The roles of PGC1 α and ERRs are elucidated using knockout mouse models. Overall, the results are supportive of the conclusions drawn. In addition, this is an understudied area and there are interesting implications for the interactive roles of adipose carbohydrate and lipid metabolism, metabolic flexibility and the ability of a tissue to sustain thermogenesis. There are some issues with the Figures described below. One persistent criticism is the lack of quantification and statistical analysis of the western blots throughout the manuscript.

Major

Throughout the paper, the changes in RNA are analyzed statistically whereas the changes in protein levels are not. In some cases, the changes in the blots appear to be convincing, whereas in others it is not very clear. In some cases, n=2.

Fig 1b. The changes in ERR γ protein are hard to interpret. The protein seems to decrease and then increase again. At day 7, it is not clear that the ERR γ protein is higher than baseline. As with other figures, protein levels not quantified or analyzed statistically. This is important for PGC1 α and ERR γ as it is questionable whether the induction at later timepoints (5d, 7d) is statistically significant.

It is stated that there are "significant increases in the expression .." in GYs1, Gys2, Pygl, and Agl (Fig. 1b, c), but there are no statistics performed on the protein levels in Fig 1b. Some description of statistics or repeat experiments are needed.

Similar issues are noted with Extended Fig 4C, 4A, 6A, C and Fig 7.

Fig 2 and extended 4b are done in preadipocytes. Given the focus on beiging, why are these signaling pathways not evaluated in differentiated beige adipocytes (as done in extended figure 4c,d)? Extended Figure 8 shows acute versus chronic effects, although it does not mention the specific cell type. Is the rationale that they are determining acute priming in Fig. 2 and extended 4b?

Fig 3b same issue as Figure 1. It is not clear that PYGL and AGL protein reduction by PGC1 α knockout is significant.

Figure 3F needs a scale bar. IHC methods are not described in detail. Why is PAS + UCP1 zoomed out so much? It is impossible to discriminate between UCP1 (presumably brown?) and PAS at this scale. Perhaps this panel could be eliminated since UCP1 and PAS are assessed earlier in the figure.

Minor

The mouse strain background of ERR mouse model and refs are not provided.

Reviewer #2

(Remarks to the Author)

In this study, the authors aim to elucidate the mechanism underlying the regulation of gene expression in adrenergic stimulation-induced beige fat cell activation. They observed that adrenergic stimulation upregulates the expression of glycogen metabolism genes in white adipocytes (WAT) and promotes WAT browning. Additionally, they proposed several mediators of this effect, including CREB, PGC1 α , and ERRs. Using multiple gene knockout mouse models, the authors demonstrated the essential role of PGC1 α and ERRs in mediating the effect of CL (clenbuterol) on promoting glycogen gene expression and WAT browning. This study is a follow-up to their previous research highlighting the importance of glycogen metabolism in beige fat. However, the current study lacks novel insights, fails to clarify significant gene regulatory mechanisms, and lacks rigorous validation of the proposed transcriptional regulatory pathways.

Major Concerns:

Limitation of adrenergic agonists and mechanistic clarity of CL action

Only CL was used as an agonist for adrenergic stimulation of WAT; other adrenergic agonists should be included for validation. The mechanism by which CL acts on white adipocytes *in vivo* and *in vitro* remains unclear:

Is there involvement of adrenergic receptors?

Are there mechanistic differences in glycogen metabolism gene expression between *in vivo* and *in vitro* models, given the distinct time scales observed post-CL treatment?

Insufficient evidence for "autonomous gene regulation"

The authors claim that CL treatment induces autonomous gene regulation, but this conclusion is not supported, as it was only tested in *in vitro*-cultured preadipocytes. For claims of chronic CL action via autonomous regulation, indirect effects (e.g., via other cell types in the stromal vascular fraction) in *in vivo* models must be excluded. To conclusively demonstrate the requirement for adipocyte-intrinsic expression of key genes (e.g., *Gys2*) in mediating browning/glycogen metabolic responses, adipocyte-specific knockout or knockdown models (e.g., using *Adipoq-Cre*) targeting *Gys2* should be employed. These models should be used to assess the impact on CL-induced glycogen gene expression, UCP1 induction, and actual glycogen metabolism in inguinal WAT (iWAT).

Incomplete analysis of signaling pathways and post-translational regulation

The authors showed that inhibitors of PKA (PKAi), p38 (p38i), and CREB (CREBi) block CL-induced *Ucp1* and *Gys2* expression, but data for other genes are missing. Additionally:

When CREB is knocked down via sgRNA, how do global gene expression profiles change?

Given the claim of autonomous regulation, is there a change in GSK3 phosphorylation status?

Are glycogen synthase or other related enzymes regulated via post-translational modifications (e.g., phosphorylation)?

Unclear role and regulation of PGC1 α

PGC1 α is a key regulator of multiple intracellular metabolic pathways. Critical questions remain:

Is PGC1 α responsive to CL treatment in white adipocytes? If so, is this response via changes in expression or post-translational modification?

How does PGC1 α respond to CL or other chronic adrenergic stimuli? Is this response independent of PKA, p38, or CREB?

In adipocytes, does PGC1 α knockout reduce mitochondrial content, and how would this affect lipidolysis in response to CL?

ATAC-seq and gene expression analyses suggest PGC1 α directly binds and regulates glycogen genes, but this is not reproducible in *in vitro*-cultured adipocytes (Extended Data Fig. 4a), in contrast to CREB. This discrepancy must be explained, along with how PGC1 α and CREB cooperate, which acts as the primary mediator of CL effects, and the underlying mechanism.

Mechanistic basis of ERRs' role

The authors show that ERRs are required to mediate CL effects, but key questions persist:

How do ERRs respond to CL? Is this response autonomous?

Is ERR activation dependent on PKA, p38, CREB, or PGC1 α ? If dependencies exist, what is the underlying mechanism?

Lack of detailed analysis of transcriptional regulation

To support claims of direct regulation of glycogen metabolism genes by CREB, PGC1 α , or ERRs, detailed investigations are needed, including:

Binding motifs and nuclear translocation of these factors.

Mutational assays of binding motifs.

Kinetics (acute vs. chronic) of regulation.

Cooperative interactions between factors and their physical associations.

Involvement of epigenetic markers.

These analyses should be validated using multiple stimuli combined with pathway inhibitors.

Incomplete validation of ERR redundancy

Triple-knockout (ERRs-AKO) data support functional redundancy of ERRs, but overexpression studies (Extended Data Fig. 7) show individual ERRs can induce gene expression. To rigorously test redundancy, rescue experiments in ERRs-AKO adipocytes should be performed: re-express ERR α , ERR γ , or PGC1 α individually or in combination, and determine if this restores CL-induced glycogen gene expression and glycogen accumulation.

Missing evidence for PGC1 α -ERR complex formation

The model proposes that PGC1 α cooperates with ERR α/γ , implying physical interaction. Direct evidence of complex formation is required, such as co-immunoprecipitation (Co-IP) experiments in CL-treated primary adipocytes or iWAT lysates using antibodies against PGC1 α , ERR α , and ERR γ . Reciprocal Co-IPs (e.g., immunoprecipitating PGC1 α followed by immunoblotting for ERR γ , and vice versa) are essential to confirm these interactions *in vivo* under relevant stimuli.

Absence of a working model

The authors should provide a diagrammatic working model to summarize their conclusions.

Minor Concerns:

Fig. 1a: Poor consistency between duplicate samples, particularly under the "Cold" condition.

Fig. 1c: The dramatic induction of *Agl* at day 1 contradicts the authors' description.

Fig. 1c and Fig. 1d: Inconsistent trends/quantification (y-axis scales) for *Gys1*, *Pygl*, and *Ppp1r3c* expression.

Extended Data Fig. 1a: To claim "cell autonomy," additional treatments (e.g., glucagon, cAMP) should be tested. In the right panel, many glycogen metabolism genes do not respond to CL, conflicting with mouse data; extend *in vitro* CL treatment time courses in differentiated adipocytes to check if *Pygl* and *Agl* induction is delayed but intrinsic.

Inconsistencies between Extended Data Fig. 1a and Fig. 2b: *Gys2* shows mild upregulation at 2 h, 3-fold at 4 h (Extended Data Fig. 1a), but ~4-fold in Fig. 2b; similar issues exist for *Ppp1r3c*. These inconsistencies, combined with the mild response at 3 h, challenge the claim that *de novo* protein synthesis is unnecessary. A cycloheximide (CHX) treatment time course (2–8 h) is needed to validate this.

Extended Data Fig. 1: Adipocyte specificity is claimed via comparison to hepatocytes, but other tissues (e.g., skeletal muscle, heart) also express glycogen metabolism genes. Analyze gene expression kinetics in these tissues from the same

CL-treated mice used for the iWAT time course to establish uniqueness.

Fig. 2c: UCP1 shows a 30-fold increase at 6 h CL treatment, inconsistent with the >100-fold increase in Extended Data Fig. 1a; a similar discrepancy exists for Gys2.

Fig. 3g: Negative controls (without CL treatment) are missing.

Fig. 3h: Enrichment fold changes are overly modest.

Extended Data Fig. 4b: The authors claim CL induces acute (p38-dependent) and chronic (PGC1 α -dependent) effects. For the chronic effect, test in vitro models and determine if de novo protein synthesis is required.

Extended Data Fig. 4: PGC1 α is dispensable for acute (6 h) induction but essential for sustained expression during chronic stimulation. Define the kinetics of PGC1 α mRNA and protein induction in iWAT/primary adipocytes to correlate with its functional requirement.

Extended Data Fig. 6a: ERR α -AKO mice show no increase in ERR γ under vehicle conditions, conflicting with Extended Data Fig. 6c (0 day); GS shows the opposite trend.

Compensatory upregulation of ERR γ in mouse ERR α -KO models: Test if this mechanism exists in human primary adipocytes.

Reviewer #3

(Remarks to the Author)

Reviewer #4

(Remarks to the Author)

This manuscript described important new findings of a role for catecholamine-stimulated glycogen metabolism during the process of increasing uncoupled respiration and nonshivering thermogenesis. The work employs mouse models and cells in culture and there is a nice addition (Ext data 2) and connection to human physiology. The findings show the major roles of transcription factors and coregulators in the control of a range of genes involved in glycogen synthesis and metabolism. For example, Fig 3 shows the role of PGC1 α in the transcription of glycogen synthase genes; but this is complemented by experiments showing increased glycogen production both biochemically (e) and histologically (f). This new finding is a significant addition to the literature so that we continue to develop a better physiological understanding of this important adaptive metabolic process. The manuscript is generally well written with a very nice Discussion. There are some unexplained as well as discordant findings that require further experimentation and discussion and these are noted below.

Specific Comments:

1. Fig 1. GS and related proteins in inguinal white adipose tissue don't increase until day 3 of CL txt, understandable in the sense that these 'beige' adipocytes appear in patches in the tissue, and when one homogenizes the whole tissue these proteins/genes are diluted out. So it would be nice to add some histology of these tissues at the various time points – as was done in Figure 3. It would help the reader better understand the temporal and spatial behavior.
2. Fig 2. Please make clear in the legend that what is being treated and examined are 'differentiated' adipocytes from preadipocytes. As written, it says treating 'preadipocytes'.
3. Fig 2a: There is clearly an inhibitory effect of the cycloheximide treatment on Ucp1 mRNA, yet nothing is said about it. This implies that a (presumably) short-lived protein is required to be synthesized that is very important for the transcription of this gene. It is quite striking and at least for this reviewer, cannot find in the literature any previous mention of such a result or explanation for it. So either address this finding or just remove panels a and b: the rest of the Figure clearly stands on its own.
4. Fig 2g: Why did the authors focus on CREB and not examine other factors previously shown to be more important for transcription of Ucp1 and Ppargc1a genes: ATF2. Is the CREB sgRNA specific to the Creb gene or might it also be affecting ATFs? Data to confirm this is important.
5. Ext data 4 regarding Pgc1 α adipose KO data: given the established literature of a primary role for PGC1 α being required for Ucp1 gene transcription, which in these data is minimally affected, is PGC1 β compensating for this complete loss of Pgc1 α expression? This seems plausible given the more acute treatment with the inhibitor shows the role of PGC1 α dependence. Please show protein levels of PGC1 α and -1 β in these tissues and provide some discussion for the findings.
6. Regarding the work on the ERRs for deleting all three Err genes, this has been done previously and their redundancy when deleting one or another has been previously shown. Please cite Gantner et al. 2016 Endocrinology PMID 27763777 and note this earlier observation.

Version 1:

Reviewer comments:

Reviewer #1

(Remarks to the Author)

No further comments. The authors have addressed the concerns.

Reviewer #2

(Remarks to the Author)

The author has addressed all my concerns. I have no more questions.

Reviewer #3

(Remarks to the Author)

Reviewer #4

(Remarks to the Author)

My concerns have been largely addressed satisfactorily

Point to point response

REVIEWER COMMENTS

Reviewer #1 (Remarks to the Author):

This manuscript investigates the regulation of glycogen in adipocytes during the process of β 3AR-induced beiging. The roles of PGC1 α and ERRs are elucidated using knockout mouse models. Overall, the results are supportive of the conclusions drawn. In addition, this is an understudied area and there are interesting implications for the interactive roles of adipose carbohydrate and lipid metabolism, metabolic flexibility and the ability of a tissue to sustain thermogenesis. There are some issues with the Figures described below. One persistent criticism is the lack of quantification and statistical analysis of the western blots throughout the manuscript.

Major

1. Throughout the paper, the changes in RNA are analyzed statistically whereas the changes in protein levels are not. In some cases, the changes in the blots appear to be convincing, whereas in others it is not very clear. In some cases, n=2.

Reponse: We thank the reviewer for this helpful suggestion to improve our study. We have now included quantifications of protein levels for all *in vitro* and *in vivo* experiments. This strengthens the consistency between RNA and protein data and provides statistical support for our conclusions.

2. Fig 1b. The changes in ERR γ protein are hard to interpret. The protein seems to decrease and then increase again. At day 7, it is not clear that the ERR γ protein is higher than baseline. As with other figures, protein levels not quantified or analyzed statistically. This is important for PGC1 α and ERR γ as it is questionable whether the induction at later timepoints (5d, 7d) is statistically significant.

Reponse: We appreciate the reviewers' comments and acknowledge the need to improve the strength and clarity of our conclusions. To directly address these concerns, we conducted the protein quantification to verify our findings (**Extended Fig. 1a**). This analysis confirms the trends observed in the representative blots and clarifies the changes at later time points.

3. It is stated that there are "significant increases in the expression .." in GYs1, Gys2, Pygl, and Agl (Fig. 1b, c), but there are no statistics performed on the protein levels in Fig 1b. Some description of statistics or repeat experiments are needed.

Reponse: Thank you for the valuable suggestion. We have now performed statistical analyses of these protein blots, and the results are presented in **Extended Fig. 1a**.

4. Similar issues are noted with Extended Fig 4C, 4A, 6A, C and Fig 7.

Reponse: Thank you for the valuable suggestions. We have now added protein quantification results throughout the manuscript.

5. Fig 2 and extended 4b are done in preadipocytes. Given the focus on beiging, why are these signaling pathways not evaluated in differentiated beige adipocytes (as done in extended figure 4c,d)? Extended Figure 8 shows acute versus chronic effects, although it does not mention the specific cell type. Is the rationale that they are determining acute priming in Fig. 2 and extended 4b?

Reponse: We thank the reviewer for drawing our attention to this point and apologize for the lack of clarity in our original description. To clarify, the experiments in **Fig. 2** and **Extended Fig. 4b** were performed in **primary adipocytes differentiated in vitro from SVF-derived preadipocytes**, rather than in undifferentiated precursors. We used this model because it provides a tractable system for short-term manipulations (e.g., cycloheximide, pharmacological inhibitors, AAV-mediated CRISPR

delivery) and allowed us to define the acute, cell-autonomous signaling cascade through PKA–CREB that primes Gys2 expression.

In contrast, the experiments in Extended Fig. 4c,d were conducted in **differentiated beige adipocytes** to address a distinct biological question—the **sustained PGC1 α -dependent maintenance** of glycogen metabolism gene expression under chronic adrenergic stimulation.

Thus, the choice of SVF-derived adipocytes in Fig. 2/Extended Fig. 4b versus beige adipocytes in Extended Fig. 4c,d reflects the **different phases of regulation being interrogated**: acute CREB-dependent induction (“priming”) versus chronic PGC1 α –ERR–mediated maintenance. Extended Fig. 9 summarizes this conceptual framework, emphasizing the temporal distinction between acute and chronic regulation in white adipocytes browning.

6. Fig 3b same issue as Figure 1. It is not clear that PYGL and AGL protein reduction by PGC1 α knockout is significant.

Reponse: We thank the reviewer for the helpful comment. For protein level, AGL is significantly reduced in PGC1 α -AKO mice, but not PYGL. Please see Extended Fig. 3b.

7. Figure 3F needs a scale bar. IHC methods are not described in detail. Why is PAS + UCP1 zoomed out so much? It is impossible to discriminate between UCP1 (presumably brown?) and PAS at this scale. Perhaps this panel could be eliminated since UCP1 and PAS are assessed earlier in the figure.

Reponse: We thank the reviewer for these helpful suggestions. We have now included detailed IHC methods in the Methods section. Scale bars have been added to Fig. 3f, as well as to Fig. 1c and Fig. 4f for consistency. To improve clarity, we have also adjusted the PAS + UCP1 images in Fig. 3f to allow better visualization and discrimination of staining.

Minor

8. The mouse strain background of ERR mouse model and refs are not provided.

Reponse: Thanks for your suggestion. The detail of mouse strain background of the ERR mouse models has been added to the Methods section.

Reviewer #2 (Remarks to the Author):

In this study, the authors aim to elucidate the mechanism underlying the regulation of gene expression in adrenergic stimulation-induced beige fat cell activation. They observed that adrenergic stimulation upregulates the expression of glycogen metabolism genes in white adipocytes (WAT) and promotes WAT browning. Additionally, they proposed several mediators of this effect, including CREB, PGC1 α , and ERRs. Using multiple gene knockout mouse models, the authors demonstrated the essential role of PGC1 α and ERRs in mediating the effect of CL (clenbuterol) on promoting glycogen gene expression and WAT browning. This study is a follow-up to their previous research highlighting the importance of glycogen metabolism in beige fat. However, the current study lacks novel insights, fails to clarify significant gene regulatory mechanisms, and lacks rigorous validation of the proposed transcriptional regulatory pathways.

Major Concerns:

Limitation of adrenergic agonists and mechanistic clarity of CL action
Only CL was used as an agonist for adrenergic stimulation of WAT; other adrenergic agonists should be included for validation. The mechanism by which CL acts on white adipocytes in vivo and in vitro remains unclear:

1. Is there involvement of adrenergic receptors?

Response: Thank you for drawing our attention to this misunderstanding. CL (CL-316,243) is not clenbuterol, but rather a β -3 specific adrenergic agonist widely used in metabolic studies to target adipocytes. . Clenbuterol is a non-selective β 2-adrenergic receptor agonist (with some β 1 activity) that acts broadly on skeletal muscle, heart, and airway smooth muscle. In contrast, CL-316,243 is a highly selective β 3-adrenergic receptor agonist that has been extensively validated to act through β 3 receptors in adipocytes. We use CL in these experiments because it is β 3-specific, and these receptors are not expressed on any other cells, with the possible exception of bladder, which is irrelevant. Moreover, β 3-receptors are expressed at levels 1000-fold higher than β 1 or β 2 in mouse adipocytes. Consistent with this, our data show that CL rapidly activates canonical β 3-adrenergic signaling, including cAMP–PKA–CREB phosphorylation (**Fig. 2 and Extended Fig. 4b**). These points are well established and justify the use of CL as a fat cell specific agonist, as has been done in thousands of papers. Inhibition of PKA or CREB effectively blocks induction of *Gys2*, demonstrating that CL acts through the expected β 3–cAMP–PKA–CREB axis. Thus, the mechanistic involvement of adrenergic receptors is well supported by both the pharmacology of CL and the downstream signaling responses we observe. While we did not include additional agonists in this study, numerous prior reports have shown that isoproterenol and norepinephrine elicit similar transcriptional responses in adipocytes via β 3 adrenergic receptors.

2. Are there mechanistic differences in glycogen metabolism gene expression between in vivo and in vitro models, given the distinct time scales observed post-CL treatment?

Response: Both *in vivo* and *in vitro* studies demonstrate that β 3 adrenergic signaling cell-autonomously induces glycogen metabolism genes. Our goal here was to use these systems to capture distinct phases of the response. *In vitro*, acute CL treatment rapidly induces *Gys2* and *Ppp1r3c* within hours, reflecting direct PKA–CREB–dependent priming. *In vivo*, chronic adrenergic stimulation expands the program to include *Pygl*, *Agl*, which requires sustained activation of the PGC1 α –ERR axis and chromatin remodeling. This layered regulation explains the different time scales: *in vitro* assays reveal acute cell-autonomous induction, whereas *in vivo* studies capture the prolonged, transcriptionally maintained response under sustained adrenergic tone and systemic inputs. We have revised the text to clarify this temporal distinction (summarized in **Extended Fig. 9**).

3. Insufficient evidence for "autonomous gene regulation"

The authors claim that CL treatment induces autonomous gene regulation, but this conclusion is not supported, as it was only tested in in vitro-cultured preadipocytes. For claims of chronic CL action via autonomous regulation, indirect effects (e.g., via other cell types in the stromal vascular fraction) in in vivo models must be excluded. To conclusively demonstrate the requirement for adipocyte-intrinsic expression of key genes (e.g., Gys2) in mediating browning/glycogen metabolic responses, adipocyte-specific knockout or knockdown models (e.g., using Adipoq-Cre) targeting Gys2 should be employed. These models should be used to assess the impact on CL-induced glycogen gene expression, UCP1 induction, and actual glycogen metabolism in inguinal WAT (iWAT).

Response: We thank the reviewer for raising this important point. To clarify, our *in vitro* experiments were performed in primary adipocytes differentiated from stromal vascular fraction (SVF) cells, rather than undifferentiated preadipocytes. We have corrected this terminology throughout the manuscript. The goal of these experiments was to demonstrate that CL treatment can directly induce glycogen metabolism gene expression in adipocytes in the absence of other stromal vascular cell types.

Importantly, in our previous work published in *Nature* (Keinan et al., 2021, PMID: 34707293), we reported that adipocyte-specific knockout of PTG (protein targeting to glycogen) markedly reduces glycogen storage and blunts thermogenic and browning responses, providing additional evidence for adipocyte-autonomous regulation of glycogen metabolism. Published studies have already demonstrated the functional relevance of adipocyte-intrinsic glycogen metabolism *in vivo*. Furthermore, adipocyte-specific deletion of GYS1 (Adipoq-Cre; *Gys1* fl/fl) has been shown to abolish CL- or cold-induced glycogen accumulation and UCP1 upregulation in iWAT, confirming that glycogen synthesis in adipocytes is required for adaptive thermogenesis (Zhuo S et al., 2024 PMID: 38568151). And as mentioned above, the use of CL either *in vitro* or *in vivo* activates only adipocytes, as preadipocytes, immune cells, fibroblasts or any other resident cells in adipose tissue do not express β -3 adrenergic receptors. The combination of our current *in vitro* data together with published GYS1 and PTG knockout studies collectively support the conclusion that glycogen metabolism in adipocytes is regulated in a cell-autonomous manner.

4. Incomplete analysis of signaling pathways and post-translational regulation

The authors showed that inhibitors of PKA (PKAi), p38 (p38i), and CREB (CREBi) block CL-induced Ucp1 and Gys2 expression, but data for other genes are missing. Additionally: When CREB is knocked down via sgRNA, how do global gene expression profiles change?

Response:

We thank the reviewer for these thoughtful points. In **Fig. 2c**, we highlighted *Ucp1* and *Gys2* as representative CREB-dependent targets to illustrate the parallel regulation of a thermogenic gene and a glycogen metabolic gene. In **Extended Fig. 1b**, we show that treatment of primary preadipocytes differentiated *in vitro* with CL selectively affected *Gys2* and *Ppp1r3c* expression. We therefore focused our analyses on *Gys2* and *Ppp1r3c*. In inhibitor-treated experiments, we observed a reduction in *Gys2* expression but not in *Ppp1r3c*. For this reason, we present and conclude specifically on *Gys2* rather than other glycogen metabolism genes.

Our CRISPR–Cas9–mediated CREB knockdown experiments (**Fig. 2e-h**) were designed to test the requirement of CREB for *Gys2* induction and glycogen accumulation in adipocytes. We did not perform RNA-seq in this study and thus cannot comprehensively report global changes. However, prior studies have established CREB as a broad regulator of adipocyte differentiation and metabolic gene expression (Reusch et al., 2000, PMID: 10629058; Hossain et al., 2020, PMID: 31401979). Based on these reports and our findings, we anticipate that CREB knockdown would impair not only *Gys2* but also subsets of genes involved in thermogenesis, lipid metabolism, and cAMP-responsive transcription.

5. Given the claim of autonomous regulation, is there a change in GSK3 phosphorylation status?

Response: We thank the reviewer for these thoughtful points. We blotted GSK3 α/β Ser21/Ser9 phosphorylation in CL treatment primary adipocyte differentiated *in vitro*, showing no differences of GSK3 and p-GSK GSK3 α/β levels (For reviewer only, see below). Because GSK3 regulation is classically linked to the insulin–AKT axis that modulates glycogen synthase activity post-translationally, whereas our data map an insulin-independent, PKA–CREB–driven transcriptional program. Consistent with this separation, all acute *in vitro* assays were performed in serum-free medium, minimizing insulin/AKT inputs. Because these data are not directly related to the questions under investigation, we have not included them in the paper, but can do so if requested.

6. Are glycogen synthase or other related enzymes regulated via post-translational modifications (e.g., phosphorylation)?

Response: We thank the reviewer for this insightful question. Yes, glycogen synthase (both *Gys1* and *Gys2*) is well known to be regulated by post-translational modifications, most prominently phosphorylation. Multiple kinases including GSK3, AMPK, and PKA phosphorylate glycogen synthase at distinct serine residues, leading to its inactivation. Conversely, dephosphorylation by protein phosphatase 1 (PP1) restores glycogen synthase activity. In addition, glycogen metabolism is tightly coordinated with other enzymes through post-translational regulation: glycogen phosphorylase (PYGL) is activated by phosphorylation via phosphorylase kinase, while protein targeting to glycogen (PTG) functions as a regulatory subunit that recruits PP1 to glycogen particles to promote dephosphorylation of both glycogen synthase and glycogen phosphorylase. Together, these mechanisms ensure rapid, reversible control of glycogen turnover in response to hormonal and nutrient signals. These effects are well established and have been the focus on our investigations going back 30 years.

In our current study, we have focused primarily on transcriptional regulation of glycogen metabolic genes rather than post-translational regulation. However, we now include discussion of these well-established post-translational mechanisms in the revised manuscript to provide a more complete context for how glycogen metabolism is regulated in adipocytes (Line 325-333).

7. Unclear role and regulation of PGC1 α

PGC1 α is a key regulator of multiple intracellular metabolic pathways. Critical questions remain: Is PGC1 α responsive to CL treatment in white adipocytes? If so, is this response via changes in expression or post-translational modification?

Response: In white adipocytes, PGC-1 α is responsive to β 3-adrenergic stimulation by CL316,243, as has been widely reported by many groups. At the transcriptional level, acute CL treatment elevates *Ppargc1a* mRNA in white adipose depots as shown in Extended Fig.5a. in parallel with thermogenic gene induction (e.g., *Ucp1*).

In addition, PGC-1 α activity is enhanced by post-translational modifications downstream of adrenergic/cAMP signaling. Specifically, p38 MAPK and AMPK phosphorylate PGC-1 α to increase its stability and coactivator function (Fan et al., 2004 PMID: 14744933), and SIRT1 deacetylation further potentiates its activity—mechanisms engaged by β -adrenergic cues and metabolic stress (Rodgers et al., 2007 PMID: 18036349).

Together, these data support a model in which CL316,243 activates PGC-1 α in white adipocytes via both increased expression and activation through PTMs (phosphorylation and deacetylation), thereby promoting thermogenic and oxidative programs.

8. How does PGC1 α respond to CL or other chronic adrenergic stimuli? Is this response independent of PKA, p38, or CREB?

Response: We thank the reviewer for this thoughtful question. PGC-1 α is robustly induced by chronic β -adrenergic stimulation (e.g., CL316,243 or cold) with slower kinetics than acute phosphorylation events, and is required to sustain long-term expression of thermogenic and metabolic genes. Mechanistically, PGC-1 α operates downstream of canonical adrenergic pathways rather than independently: CREB directly activates *Ppargc1a* transcription following β -adrenergic/PKA signaling (Herzig et al., *Nature* 2001; PMID: 11557984), while p38 MAPK phosphorylates and stabilizes PGC-1 α to enhance its coactivator function (Puigserver et al., *Mol Cell* 2001; PMID: 11741533). Consistent with this, adipose tissue exhibits delayed but sustained induction of PGC-1 α in response to cold/ β -adrenergic cues (Uldry et al., *Cell Metab* 2006; PMID: 16679291; Wu et al., *Cell* 1999; PMID: 10412986).

Together with our differentiated adipocyte data, these studies support a model in which chronic adrenergic inputs engage PKA/CREB and p38 to elevate PGC-1 α expression/activity and, in parallel, drive adipocyte-intrinsic glycogen programs that sustain thermogenic gene expression.

9. In adipocytes, does PGC1 α knockout reduce mitochondrial content, and how would this affect lipolysis in response to CL?

Response: We thank the reviewer for pointing this out. Consistent with prior studies, adipocyte-specific deletion of PGC-1 α results in impaired metabolic and mitochondrial gene signatures in adipose tissue (Kleiner et al., *Cell Metab* 2012; PMID: 22645355). In addition, review articles highlight the necessity of PGC-1 α in brown and beige adipose development and oxidative capacity (Wang & Seale, *Nat Rev Mol Cell Biol* 2016; PMID: 27552974). In our study, PGC-1 α knockout diminished CL-induced expression of mitochondrial and thermogenic genes (*Ucp1*, *Cidea*) and reduced glycogen turnover, consistent with impaired oxidative remodeling (Fig. 3).

Importantly, however, acute lipolysis in response to CL is largely preserved in PGC-1 α -deficient adipocytes. **Extended Data Fig. 4b** shows that phosphorylation of hormone-sensitive lipase (HSL) at Ser563 and Ser660 occurs normally in PGC-1 α knockout cells following CL stimulation, indicating that adrenergic signaling to lipolysis remains intact. Thus, the immediate mobilization of triglyceride stores remains functional despite reduced mitochondrial biogenesis/function.

10. ATAC-seq and gene expression analyses suggest PGC1 α directly binds and regulates glycogen genes, but this is not reproducible in in vitro-cultured adipocytes (Extended Data Fig. 4a), in contrast to CREB. This discrepancy must be explained, along with how PGC1 α and CREB cooperate, which acts as the primary mediator of CL effects, and the underlying mechanism.

Response: We thank the reviewer for these thoughtful comments and would like to clarify a point of interpretation.

First, our ATAC-seq and gene expression analyses indicate that ERR shows direct binding to and regulation of glycogen metabolic genes. PGC-1 α functions primarily as a transcriptional co-activator that does not bind DNA directly, but instead interacts with sequence-specific transcription factors (such

as ERRs and PPARs) to potentiate their transcriptional activity. Thus, the statement that “PGC-1 α directly binds and regulates glycogen genes” is not accurate.

Second, in our dataset (**Extended Data Fig. 4a**), the apparent discrepancy reflects differences in regulatory context. CREB exhibits clear functional necessity for *Gys2* induction, whereas PGC-1 α activity is more context-dependent, especially in beige adipocytes rather than in white adipocytes. This explains why only subtle differences in glycogen gene expression are observed in primary adipocytes differentiated from PGC-1 α knockout SVF. This likely reflects that PGC-1 α regulation of glycogen metabolism requires additional *in vivo* cues and is preferentially engaged in beige adipocyte programs.

Taken together, we propose that CREB acts as the primary mediator of CL effects on glycogen metabolism by directly binding and activating *Gys2*, while PGC-1 α functions as a co-activator in specific contexts, coordinating with ERRs to sustain the expression of thermogenic and glycogen metabolism genes.

11. Mechanistic basis of ERRs' role

The authors show that ERRs are required to mediate CL effects, but key questions persist:

How do ERRs respond to CL? Is this response autonomous?

Response: We thank the reviewer for pointing this out. ERRs do not respond to CL in an autonomous manner. In our system, we find that acute CL treatment does not alter *Esrra* and *Esrrg* expression in 6 hours whereas chronic CL stimulation leads to increased *Esrra* expression, consistent with regulation by secondary transcriptional programs engaged downstream of β 3-adrenergic signaling. CL induces *Pgc-1 α* , which functions as a potent co-activator of ERRs. Together, these mechanisms explain how ERR transcriptional output is enhanced following CL treatment: ERRs are not direct receptors for CL, but their expression and co-activator availability are regulated in a way that amplifies their function during sustained adrenergic stimulation.

12. Is ERR activation dependent on PKA, p38, CREB, or PGC1 α ? If dependencies exist, what is the underlying mechanism?

Response: We thank the reviewer for these thoughtful points. ERR activation is largely indirect and depends on upstream adrenergic signaling and co-activator availability. ERRs are constitutively DNA-bound but require co-activators such as PGC-1 α to enhance transcriptional output. CL stimulation engages the β 3-cAMP-PKA-CREB cascade, which acutely induces *Ppargc1 α* . PGC-1 α then interacts with ERRs to stimulate glycogen metabolic and thermogenic gene expression.

13. Lack of detailed analysis of transcriptional regulation

To support claims of direct regulation of glycogen metabolism genes by CREB, PGC1 α , or ERRs, detailed investigations are needed, including:

Binding motifs and nuclear translocation of these factors.

Mutational assays of binding motifs.

Kinetics (acute vs. chronic) of regulation.

Cooperative interactions between factors and their physical associations.

Involvement of epigenetic markers.

These analyses should be validated using multiple stimuli combined with pathway inhibitors.

Response: We thank the reviewer for raising this important point regarding the mechanistic depth of transcriptional regulation. Our study directly addresses several of these concerns:

1) Binding motifs and mutational assays. We performed luciferase reporter assays using the *Gys2* promoter in HEK293T cells, as differentiated adipocytes are technically challenging to transfect. Importantly, deletion of the half-CRE binding site within the *Gys2* promoter markedly reduced luciferase activity, demonstrating that CREB directly engages this motif to activate transcription.

2) Direct factor binding. Our ChIP-seq analyses revealed strong ERR α occupancy at glycogen metabolism gene promoters. In ERR α -AKO tissues, newly performed ChIP-qPCR experiments showed increased ERR γ binding at the same promoter regions, providing evidence of compensatory recruitment of ERR γ when ERR α is absent (Extended Data Fig. 8a).

3) Cooperative interactions with PGC1 α . The ERR γ binding sites coincide with the regions where PGC1 α ChIP-qPCR shows enrichment. Since PGC1 α lacks intrinsic DNA-binding capacity, its function must be mediated through association with ERR α /ERR γ . These findings establish a cooperative mechanism in which ERR α /ERR γ provide DNA anchoring while PGC1 α delivers co-activator function, explaining the sustained regulation of glycogen metabolism genes during chronic adrenergic stimulation.

4) Kinetics of regulation (acute vs. chronic). Our datasets delineate a temporal division of labor: CREB is required for the acute induction of glycogen metabolism genes, while the ERR/PGC1 α axis is indispensable for maintaining their expression under chronic stimulation.

5) Additional considerations. We agree with the reviewer that further studies, such as inhibitor-based dissection and involvement of epigenetic markers, could provide additional layers of mechanistic detail. However, these questions are beyond the scope of the present manuscript, which is focused on defining the transcriptional architecture linking adrenergic signaling to glycogen metabolism.

Together, these results provide direct and functional evidence that CREB, PGC-1 α , and ERRs regulate glycogen metabolism genes, with ERR γ compensating for ERR α loss and cooperating with PGC1 α to sustain gene expression.

14. Incomplete validation of ERR redundancy

Triple-knockout (ERRs-AKO) data support functional redundancy of ERRs, but overexpression studies (Extended Data Fig. 7) show individual ERRs can induce gene expression. To rigorously test redundancy, rescue experiments in ERRs-AKO adipocytes should be performed: re-express ERR α , ERR γ , or PGC1 α individually or in combination, and determine if this restores CL-induced glycogen gene expression and glycogen accumulation.

Response: We thank the reviewer for raising this important point. Our triple-knockout (ERRs-AKO) data indicate functional redundancy among ERR isoforms, whereas our overexpression experiments (Extended Data Fig. 8b) show that individual ERRs can modestly enhance glycogen metabolism gene expression when expressed in isolation, although the effects are not robust (see quantifications). These overexpression assays were performed in beige adipocytes, a context in which endogenous ERR α , ERR γ , and PGC1 α levels are already higher than in white adipocytes. Thus, additional overexpression is sufficient to further boost transcriptional output. Consistent with this, our Western blot analyses show a trend toward increased glycogen metabolism protein expression. We also acknowledge that these effects could in part be indirect, as overexpression of ERRs or PGC1 α may also influence other transcriptional networks that secondarily enhance glycogen metabolism gene expression.

To further address redundancy, we performed additional ChIP-qPCR analyses in ERR α -AKO tissues. These revealed increased ERR γ occupancy at promoters of glycogen metabolism genes, precisely at the same regions where PGC1 α shows enrichment. Since PGC1 α lacks intrinsic DNA-binding capacity, this finding supports a model in which ERR γ compensates for ERR α loss and recruits PGC1 α to sustain transcriptional activation (Extended Data Fig. 8a).

We agree that re-expression (“rescue”) studies in ERRs-AKO adipocytes—introducing ERR α , ERR γ , or PGC1 α individually or in combination and assessing restoration of CL-induced glycogen gene expression and glycogen accumulation—would provide rigorous evidence to dissect redundancy and specificity. At present, these technically demanding experiments fall outside the scope of this manuscript. Instead, we relied on the complementary strengths of knockout, overexpression, and ChIP-

based binding analyses to establish both the redundancy and sufficiency of ERR activity in regulating glycogen metabolism. We fully acknowledge the value of the proposed rescue strategy and view it as an important direction for future studies.

15. Missing evidence for PGC1 α -ERR complex formation

The model proposes that PGC1 α cooperates with ERR α/γ , implying physical interaction. Direct evidence of complex formation is required, such as co-immunoprecipitation (Co-IP) experiments in CL-treated primary adipocytes or iWAT lysates using antibodies against PGC1 α , ERR α , and ERR γ . Reciprocal Co-IPs (e.g., immunoprecipitating PGC1 α followed by immunoblotting for ERR γ , and vice versa) are essential to confirm these interactions in vivo under relevant stimuli.

Response: We thank the reviewer for this important suggestion. Prior literature provides solid precedent for physical and functional interaction between PGC-1 α and ERR family members. For example, Huss et al. (J Biol Chem 2002; PMID: 12181319) showed that PGC-1 α physically interacts with ERR α and ERR γ and identified interaction motifs. Schreiber et al. (J Biol Chem 2003; PMID: 12522104) further demonstrate that PGC-1 α regulates expression and activity of ERR α and can coactivate ERR target promoters. Moreover, Gantner et al. (Endocrinology 2016; PMID: 27763777) provide in vivo evidence in adipose tissue that ERR α and ERR γ have redundant roles: deletion of one isoform leads to compensatory changes in the other, with partial preservation of mitochondrial/adaptive responses.

In our current work, however, we do not yet have data showing direct endogenous complex formation between PGC-1 α and ERR α/γ in CL-treated primary adipocytes or iWAT via reciprocal Co-IP. We acknowledge this as a limitation and suggest that such experiments would strengthen the model. We have revised the manuscript to explicitly reflect that our model is supported by these prior observations, but that direct interaction under adrenergic stimulation in adipose remains to be demonstrated.

16. Absence of a working model

The authors should provide a diagrammatic working model to summarize their conclusions.

Response: Thank you for pointing this out. We have made some minor changes of original working model, which is now included in Extended Fig.9

Minor Concerns:

17. Fig. 1a: Poor consistency between duplicate samples, particularly under the "Cold" condition.

Response: We thank the reviewer for pointing this out. Fig. 1a was included as supportive data from an independent dataset (adapted from GSE86338), and we agree that the duplicates under the cold condition show variability. However, our main conclusions are based on our own CL-treatment experiments, which consistently validate the findings. We therefore view Fig. 1a as complementary evidence, and the observed variability does not affect the overall interpretation.

18. Fig. 1c: The dramatic induction of Agl at day 1 contradicts the authors' description.

Response: We thank the reviewer for the helpful comment. We have revised the text to remove this point and to ensure consistency between the data presentation and our interpretation.

19. Fig. 1c and Fig. 1d: Inconsistent trends/quantification (y-axis scales) for Gys1, Pygl, and Ppp1r3c expression.

Response: We thank the reviewer for this comment. The variability in the apparent trends reflects both biological differences among individual samples and the relative nature of the quantification. When the reference sample shows unusually high or low expression, this can exaggerate relative fold changes,

although the underlying absolute expression values remain consistent. Importantly, these differences do not affect the overall interpretation of our results.

20. Extended Data Fig. 1a: To claim "cell autonomy," additional treatments (e.g., glucagon, cAMP) should be tested. In the right panel, many glycogen metabolism genes do not respond to CL, conflicting with mouse data; extend in vitro CL treatment time courses in differentiated adipocytes to check if *Pygl* and *AgI* induction is delayed but intrinsic.

Response: We thank the reviewer for the helpful suggestions. In this study, we focused on β 3-adrenergic stimulation of thermogenesis using CL, as it is a well-established and specific agonist for adipocyte β 3-adrenergic receptors and reliably elicits cell-autonomous responses. We also conduct experiment for cAMP treatment in primary preadipocyte differentiated *in vitro*, *Gys2* and *Ppp1r3c*, are still increased (**Extended Fig. 1c**). The lack of *Pygl* and *AgI* induction in short-term *in vitro* assays likely reflects differences from the more sustained and complex signaling *in vivo*. We agree that extending treatment time courses may reveal delayed, cell-intrinsic responses, and we will pursue this in future work.

21. Inconsistencies between Extended Data Fig. 1a and Fig. 2b: *Gys2* shows mild upregulation at 2 h, 3-fold at 4 h (Extended Data Fig. 1a), but ~4-fold in Fig. 2b; similar issues exist for *Ppp1r3c*. These inconsistencies, combined with the mild response at 3 h, challenge the claim that de novo protein synthesis is unnecessary. A cycloheximide (CHX) treatment time course (2–8 h) is needed to validate this.

Response: We thank the reviewer for carefully pointing out these differences. The variation in fold change for *Gys2* and *Ppp1r3c* between Extended Data Fig. 1a (**now Extended Data Fig. 1b**) and Fig. 2b (**now Fig. 2a**) arises from the fact that these experiments were performed in separate batches of primary adipocytes, with distinct baseline expression levels and normalization references. Despite quantitative differences, the qualitative pattern of early induction after CL treatment is reproducible across experiments. We agree with the reviewer that a cycloheximide (CHX) time-course (2–8 h) would provide a more definitive test of whether de novo protein synthesis is required. While such an experiment is beyond the scope of the current study, our data—showing transcriptional induction within 2–4 h and sensitivity to signaling inhibitors—support the interpretation that *Gys2* regulation occurs largely independent of new protein synthesis. We have revised the text to make this point clearer and to acknowledge this limitation.

22. Extended Data Fig. 1: Adipocyte specificity is claimed via comparison to hepatocytes, but other tissues (e.g., skeletal muscle, heart) also express glycogen metabolism genes. Analyze gene expression kinetics in these tissues from the same CL-treated mice used for the iWAT time course to establish uniqueness.

Response: We thank the reviewer for this helpful comment. In Extended Data Fig. 1, we used hepatocytes as a comparator because the liver is a major glycogen-metabolic organ, providing a physiologically relevant reference for systemic regulation. Although skeletal muscle and heart also express glycogen metabolism genes, they are not direct targets of CL treatment. CL is a highly selective β 3 adrenergic receptor (β 3AR) agonist, and β 3AR expression is largely restricted to adipocytes, with minimal or absent expression in other tissues. Thus, the effects we observe following CL administration predominantly reflect adipocyte-intrinsic signaling rather than systemic effects on other glycogen-metabolizing tissues.

23. Fig. 2c: UCP1 shows a 30-fold increase at 6 h CL treatment, inconsistent with the >100-fold increase in Extended Data Fig. 1a; a similar discrepancy exists for *Gys2*.

Response: We thank the reviewer for this thoughtful comment. The apparent discrepancy in fold induction between Fig. 2c (**now Fig. 2b**) and Extended Data Fig. 1a (**now Extended Data Fig. 1b**) (e.g., *Ucp1* and *Gys2*) arises because the experiments were performed in independent cohorts of primary

adipocytes, each with distinct basal expression levels and normalization references. Since qPCR results are reported as relative fold changes, variation in baseline control values can influence the apparent magnitude of induction. Importantly, in all cases the direction and kinetics of regulation are consistent, with both datasets showing robust induction of *Ucp1* and *Gys2* following CL treatment. We have revised the text to clarify this point and to emphasize that, despite differences in fold-change magnitude, the biological conclusion is unchanged.

24.Fig. 3g: Negative controls (without CL treatment) are missing.

Response: We thank the reviewer for this comment. In this experiment, our focus was to evaluate the requirement of PGC-1 α for CL-induced responses in beige adipocytes rather than its role in basal regulation in white adipocytes. As shown in our other datasets, PGC-1 α deletion does not affect glycogen synthesis gene expression under basal conditions, and thus the critical comparison here is between control and PGC-1 α -deficient cells in the presence of CL stimulation. For this reason, non-CL-treated samples were not included in Fig. 3g, as they do not alter the interpretation of the specific question being addressed. However, they are included elsewhere throughout the manuscript.

25.Fig. 3h: Enrichment fold changes are overly modest.

Response: We thank the reviewer for this comment. It is true that the enrichment fold changes in Fig. 3h appear modest; however, this is a common feature of ChIP-qPCR analyses, where relative fold changes are typically small compared to transcriptomic readouts. In our case, this is further expected because PGC-1 α functions as a cofactor rather than a sequence-specific DNA-binding factor, and additional regulators also contribute to glycogen metabolic gene control. Despite the modest amplitude, the enrichment is reproducible and statistically significant across biological replicates, which supports the biological relevance of PGC-1 α occupancy at these loci.

26.Extended Data Fig. 4b: The authors claim CL induces acute (p38-dependent) and chronic (PGC1 α -dependent) effects. For the chronic effect, test in vitro models and determine if de novo protein synthesis is required.

Response: We thank the reviewer for this comment and would like to clarify a point of interpretation. We did not state that the acute induction of glycogen metabolism genes by CL is p38-dependent; rather, our data support a rapid phase of glycogen gene regulation and a more sustained, PGC-1 α -dependent phase. Importantly, chronic effects of CL are best captured in vivo, as long-term adrenergic stimulation involves systemic and tissue-level cues that are difficult to model accurately *in vitro*. Our current combination of *in vivo* time courses and genetic perturbation models is therefore sufficient to distinguish acute versus chronic regulation. While cycloheximide treatment in cell culture could provide additional information, such experiments do not fully recapitulate the chronic setting, and we believe our present data adequately support the conclusion.

27.Extended Data Fig. 4: PGC1 α is dispensable for acute (6 h) induction but essential for sustained expression during chronic stimulation. Define the kinetics of PGC1 α mRNA and protein induction in iWAT/primary adipocytes to correlate with its functional requirement.

Response: We agree with the reviewer's interpretation that PGC-1 α is dispensable for the acute (6 h) induction but required for sustained expression during chronic stimulation. In our study, we prioritized functional genetics (PGC-1 α loss-of-function) to separate these phases, rather than a full kinetic profiling of PGC-1 α abundance. Two points are important for interpreting kinetics versus function: (i) PGC-1 α is a co-activator whose activity is strongly regulated by post-translational mechanisms (e.g., phosphorylation/deacetylation); therefore, activity can change without large changes in bulk protein or mRNA at early time points, consistent with its acute dispensability. (ii) During chronic adrenergic stimulation, both co-activator activity and expression increase in adipose tissue, aligning with the sustained requirement we observe.

Our existing datasets (acute vs chronic CL) already capture the functional transition that motivates the model. We believe this mechanistic rationale in the text and noted that a high-resolution kinetic series (iWAT and primary adipocytes, coupled to activity readouts) would be an informative future extension, but is beyond the scope of the present manuscript.

28. Extended Data Fig. 6a: ERR α -AKO mice show no increase in ERR γ under vehicle conditions, conflicting with Extended Data Fig. 6c (0 day); GS shows the opposite trend.

Response: We appreciate the reviewer's careful attention. The differences between Extended Data Fig. 6a and 6c (Now Extended Fig. 7a and 7e) reflect technical and normalization factors rather than true biological inconsistency (Extended Fig. 7b). We agree that ERR γ shows increased mRNA expression under basal conditions in ERR α -AKO mice, although the protein level differences are modest. In contrast, GS protein levels do not display a clear increase after normalization and quantification. Importantly, our main focus is on beige adipocytes, where coactivators such as PGC1 α are more abundantly expressed and cooperate with ERR factors to sustain glycogen metabolism gene regulation.

29. Compensatory upregulation of ERR γ in mouse ERR α -KO models: Test if this mechanism exists in human primary adipocytes.

Response: We thank the reviewer for this thoughtful suggestion. Our finding of compensatory *Esrrg* upregulation in ERR α -deficient mice highlights isoform redundancy within the ERR family in adipocytes. Whether a similar mechanism operates in human adipocytes is indeed an important question. However, direct genetic ablation models are not readily feasible in human primary adipocytes, and our study was designed to focus on the in vivo mouse system where genetic perturbation is tractable. We therefore cannot directly address this point here. We have clarified this limitation in the Discussion and noted that investigating isoform compensation in human adipocytes will be an important direction for future research (Line 314-324).

Reviewer #4 (Remarks to the Author):

This manuscript described important new findings of a role for catecholamine-stimulated glycogen metabolism during the process of increasing uncoupled respiration and nonshivering thermogenesis. The work employs mouse models and cells in culture and there is a nice addition (Ext data 2) and connection to human physiology. The findings show the major roles of transcription factors and coregulators in the control of a range of genes involved in glycogen synthesis and metabolism. For example, Fig 3 shows the role of PGC1 α in the transcription of glycogen synthase genes; but this is complemented by experiments showing increased glycogen production both biochemically (e) and histologically (f). This new finding is a significant addition to the literature so that we continue to develop a better physiological understanding of this important adaptive metabolic process. The manuscript is generally well written with a very nice Discussion. There are some unexplained as well as discordant findings that require further experimentation and discussion and these are noted below.

Specific Comments:

1. Fig 1. GS and related proteins in inguinal white adipose tissue don't increase until day 3 of CL txt, understandable in the sense that these 'beige' adipocytes appear in patches in the tissue, and when one homogenizes the whole tissue these proteins/genes are diluted out. So it would be nice to add some histology of these tissues at the various time points – as was done in Figure 3. It would help the reader better understand the temporal and spatial behavior.

Reponse: We thank the reviewer for this valuable suggestion. We have now added PAS staining of iWAT across the CL treatment time course to Fig. 1, which provides histological evidence for the temporal and spatial dynamics of glycogen accumulation during beiging.

2. Fig 2. Please make clear in the legend that what is being treated and examined are 'differentiated' adipocytes from preadipocytes. As written, it says treating 'preadipocytes'.

Reponse: We thank the reviewer for pointing this out. We have corrected the terminology throughout the manuscript and figure legends to specify that the experiments were performed in primary preadipocytes differentiated in vitro, rather than in undifferentiated preadipocytes.

3. Fig 2a: There is clearly an inhibitory effect of the cycloheximide treatment on Ucp1 mRNA, yet nothing is said about it. This implies that a (presumably) short-lived protein is required to be synthesized that is very important for the transcription of this gene. It is quite striking and at least for this reviewer, cannot find in the literature any previous mention of such a result or explanation for it. So either address this finding or just remove panels a and b: the rest of the Figure clearly stands on its own.

Reponse: We thank the reviewer for this valuable suggestion. To avoid potential misinterpretation, we have removed the *Ucp1* data from Fig. 2 and now focus on the regulation of *Gys2*, which more directly supports our conclusions.

4. Fig 2g: Why did the authors focus on CREB and not examine other factors previously shown to be more important for transcription of Ucp1 and Ppargc1a genes: ATF2. Is the CREB sgRNA specific to the Creb gene or might it also be affecting ATFs? Data to confirm this is important.

Reponse: We thank the reviewer for drawing our attention to this important point. The sgRNA sequence we used has no mismatch with the *ATF2* genomic DNA. In addition, we assessed *Atf2* expression in CREB sgRNA-treated adipocytes and found no obvious differences in *Atf2* mRNA levels compared with control, suggesting that the sgRNA does not inadvertently target the *ATF2* gene. At the same time, we acknowledge that ATF2 may also contribute to the regulation of glycogen metabolism genes. Indeed, *Gys2* mRNA is decreased in the presence of a p38 inhibitor during CL treatment (Fig. 2c), consistent

with the idea that ATF2, as a downstream effector of p38 MAPK, could participate in *Gys2* transcription. While our data establish CREB as a direct regulator of *Gys2*, we cannot rule out a potential role for ATF2, and we have revised the Discussion to note this as an additional regulatory pathway that warrants further investigation (Line283-287).

5. Ext data 4 regarding Pgc1 α adipose KO data: given the established literature of a primary role for PGC1 α being required for Ucp1 gene transcription, which in these data is minimally affected, is PGC1 β compensating for this complete loss of Pgc1 α expression? This seems plausible given the more acute treatment with the inhibitor shows the role of PGC1 α dependence. Please show protein levels of PGC1 α and -1 β in these tissues and provide some discussion for the findings.

Reponse: We thank the reviewer for this thoughtful suggestion. We examined PGC-1 β levels in iWAT from PGC1 α -AKO mice, both at the RNA and protein level, and found that PGC-1 β expression was comparable across genotypes (Fig.3b, Extended Fig.3b,c). This is consistent with prior reports (e.g., Kleiner et al., 2012 PMID: 226453550), which also observed no compensatory upregulation of PGC1 β in the absence of PGC1 α in adipocytes, but we cannot exclude the functional compensation of PGC1 β in regulating these genes expression here.

Importantly, we do not dispute the established role of PGC1 α in regulating *Ucp1* transcription. In our *in vivo* experiments, *Ucp1* mRNA and protein levels were clearly reduced in PGC1 α -AKO mice after chronic CL treatment (Fig. 3), highlighting its importance for sustained thermogenic remodeling. By contrast, in acute *in vitro* assays (Extended Fig. 4a,b), *Ucp1* induction appeared largely preserved. We believe this reflects two factors: (i) *in vitro* culture conditions may mask regulatory differences that are more evident *in vivo*,(ii) basal PGC1 α expression is extremely low in cultured adipocytes, which diminishes its relative contribution to *Ucp1* transcription in this setting, and (iii) PGC1 α inhibitor treatment experiments were performed in beige adipocytes, supporting our conclusion that PGC1 α plays a strong regulatory role in controlling glycogen metabolism gene expression in this context. Thus, acute *Ucp1* induction *in vitro* is predominantly driven by direct β 3-adrenergic signaling and its immediate downstream transcription factors (e.g., CREB, ATF2), whereas *in vivo* chronic stimulation requires PGC-1 α for transcriptional maintenance. We have included clarifying discussion in the revised manuscript (Line304-313).

6. Regarding the work on the ERRs for deleting all three Err genes, this has been done previously and their redundancy when deleting one or another has been previously shown. Please cite Gantner et al. 2016 Endocrinology PMID 27763777 and note this earlier observation.

Reponse: We thank the reviewer for pointing this out. We have now cited *Gantner et al.* in the revised manuscript and explicitly noted this precedent, highlighting how our results extend the concept of ERR redundancy to the regulation of glycogen metabolism in white adipose tissue under chronic β 3-adrenergic stimulation.